# Targeting Molecular Mechanisms Underlying Treatment Efficacy and Resistance in Osteosarcoma: A Review of Current and Future Strategies

**DOI:** 10.3390/ijms21186885

**Published:** 2020-09-19

**Authors:** Ingrid Lilienthal, Nikolas Herold

**Affiliations:** 1Division of Paediatric Oncology, Department of Women’s and Children’s Health, Karolinska Institutet, SE-171 76 Stockholm, Sweden; 2Paediatric Oncology, Astrid Lindgren’s Children Hospital, Karolinska University Hospital, SE-171 76 Stockholm, Sweden

**Keywords:** osteosarcoma, chemoresistance, combination chemotherapy, drug synergy, methotrexate, doxorubicin, cisplatin, ifosfamide, tumour microenvironment, immunotherapy

## Abstract

Osteosarcoma is the most common primary malignant bone tumour in children and adolescents. Due to micrometastatic spread, radical surgery alone rarely results in cure. Introduction of combination chemotherapy in the 1970s, however, dramatically increased overall survival rates from 20% to approximately 70%. Unfortunately, large clinical trials aiming to intensify treatment in the past decades have failed to achieve higher cure rates. In this review, we revisit how the heterogenous nature of osteosarcoma as well as acquired and intrinsic resistance to chemotherapy can account for stagnation in therapy improvement. We summarise current osteosarcoma treatment strategies focusing on molecular determinants of treatment susceptibility and resistance. Understanding therapy susceptibility and resistance provides a basis for rational therapy betterment for both identifying patients that might be cured with less toxic interventions and targeting resistance mechanisms to sensitise resistant osteosarcoma to conventional therapies.

## 1. Introduction

Osteosarcoma is a rare bone-forming tumour that predominantly affects adolescents [1,2,3]. While only accounting for approximately 5% of childhood and adolescent cancers, it contributes substantially to paediatric cancer mortality, in particular, due to a failure in increasing survival as achieved for other paediatric cancers since the 1970s [4,5,6]. A minor subset of low-grade osteosarcoma can be cured by surgery alone [7], but the focus of this review is on the more common, more aggressive high-grade osteosarcoma [8]. Considered a chemoresistant tumour entity, this disease was treated primarily with surgery up until the 1970s. Even when complete local control with wide margins of the affected area was possible, 80–90% of patients ultimately suffered from fatal lung metastases within 1 year of diagnosis [9]. This demonstrates the malignant nature of osteosarcoma with the presumed presence of undetectable micrometastases at diagnosis and indicates the need for systemic treatment in addition to surgery.

The advent of chemotherapy in the second half of the 20th century allowed for the implementation of neoadjuvant and adjuvant chemotherapy in the 1970s and 1980s. This improved overall survival in osteosarcoma patients with nonmetastatic disease at diagnosis to 70% [10,11,12]. This did not, however, dramatically improve the outcome for patients harbouring macrometastases at diagnosis or those with relapsed disease, with both groups exhibiting survival rates of only 20% [1,12].

Unlike for many other paediatric cancers [5], empirical treatment intensification for osteosarcoma has failed to substantially improve survival rates during the last four decades. It therefore seems paramount to unravel the underlying mechanisms that determine chemotherapy susceptibility and resistance in osteosarcoma. This will aid in the development of tailored treatment regimens that allow both treatment de-escalation for susceptible osteosarcoma and treatment improvement for resistant osteosarcoma by targeting mechanisms of drug resistance. As osteosarcoma is a complex disease interacting with its microenvironment and the immune system, those aspects cannot be ignored when discussing osteosarcoma therapy, not the least because chemotherapy also affects the host. Hence, in the following sections, we will outline and focus on what is known about general chemotherapy resistance, the pathogenesis of osteosarcoma, current treatment strategies, and mechanisms underlying treatment resistance and susceptibility in osteosarcoma and how these can be targeted in future therapies. This will necessarily be complemented by a consideration of the osteosarcoma microenvironment and osteosarcoma immunology.

## 2. Chemotherapy Resistance

Cancer research has evolved rapidly over the past decades, leading to treatments that have improved survival rates for most cancer types [13]. While novel treatment modalities like targeted and immunotherapy have begun to transform outcomes for some subtypes of leukaemia, chemotherapy is still the regimen of choice for systemic treatment of almost all cancer types [14]. Despite the wide array of chemotherapy options available today, drug resistance is a significant problem for the majority of patients and accounts for 90% of treatment failure [14,15]. Resistance to chemotherapy can be pre-existing (intrinsic) or drug-induced (acquired) (Figure 1) [16]. Up to half of cancers exhibit intrinsic resistance, while 50% of the remaining half will acquire resistance as a result of treatment [17], even though deeper sequencing strategies at diagnosis might reveal that many presumed acquired resistances indeed was present in subclones already at diagnosis as demonstrated for leukaemia [18]. Both types of resistance are the result of altered drug metabolism and/or modified drug targets (reviewed in [15]). Osteosarcoma is a particularly chemotherapy-resistant tumour. These tumours only respond to high doses of chemotherapy and acquire resistance rapidly, reflected in the poor salvage rates of only 20% in patients with relapsing disease [1].

## 3. Molecular Biology of Osteosarcoma

Osteosarcoma is a heterogenic disease characterized by a high level of genomic instability [19], a wide range of genetic aberrations [20], and multiple disturbed signalling pathways [21]. Factors involved in its evolution and maintenance have been pinpointed to stem from pathways involved in bone development and regulation, tumour microenvironment, genomic stability and cell cycle control, and cell signalling. These molecular mechanisms will be discussed in the following sections.

### 3.1. Bone Regulation

Paediatric osteosarcoma has a peak incidence around the time of puberty, with tumours commonly found near the metaphyseal growth plate of long bones [1]. In addition, adults with Paget’s disease of bone, an inflammatory condition characterised by extensive bone remodelling, are 100 times more likely to develop osteosarcoma than the general population [22]. This suggests an aetiological link between osteosarcoma and dysregulated bone growth. Bone development depends on the balance between cells that promote bone formation, osteoblasts, and those which promote bone remodelling via resorption, osteoclasts. These cells derive from two different cell lineages: osteoblasts, like fibroblasts and myoblasts, develop from mesenchymal stem cells (MSCs), whereas osteoclasts, like macrophages, develop from circulating monocytes [21]. Proper communication between these different cell types is crucial for normal bone homoeostasis and is tightly regulated by cytokines and growth factors.

Osteoclastogenesis is promoted by cofactors such as macrophage-colony stimulating factor (M-CSF) that allow for the proliferation and survival of osteoclast precursors [23]. Upon activation, osteoclasts express receptor activator of nuclear factor κ B (RANK) [24]. Following RANK expression, further differentiation and activation can be promoted by a number of cues. Osteoblasts secrete two of the most important signals, RANK ligand (RANKL) and osteoprotegerin (OPG). Although RANKL promotes osteoclast differentiation and activation, OPG counterbalances this function by binding to RANKL and interrupting its association with RANK [25]. In this way, osteoblasts can both up- and downregulate osteoclast activity [25]. Other factors involved in osteoclast differentiation are hormones such as insulin-like growth factors (IGFs) and parathyroid hormone (PTH) [21,26,27]. Osteosarcoma mouse models exhibit reduction in metastatic disease and an 85% reduction in IGF-1 levels following removal of the pituitary gland, which suggests that IGF-1, as a downstream effector of growth hormone (GH), may be partially responsible for osteosarcoma proliferation [28]. Expression of IGF receptor 1 (*IGF1R*), *IGF-1*, and *IGF-2* have been reported in osteosarcoma cell lines and patient samples [29], further suggesting its role in osteosarcoma pathogenesis. Furthermore, children treated with GH have been found to have an increased incidence of osteosarcoma [30].

Cancers that frequently metastasise to bone release factors promoting osteoclastogenesis, indicating that osteoclasts might be a central regulator in the formation of a microenvironment favouring metastasis [31]. Most osteosarcomas are osteolytic, suggesting that osteoclasts are also involved in osteosarcoma pathophysiology [21]. However, the exact role of osteoclast in the tumour microenvironment is controversial: one study found that increased osteoclast activity was associated with higher metastatic potential [32], whereas another found that decreased osteoclast presence was associated with metastatic disease [33]. These contradictory results might be resolved by a model favouring a dynamic tumour microenvironment: initially, osteoclasts contribute to a growth-promoting niche, while tumour heterogenicity at later stages leads to phenotypes that inhibit osteoclastogenesis, which destroys this niche and promotes tumour cell migration and metastasis [31].

Malignant osteosarcoma cells share characteristics of immature osteoblasts, suggesting that an aberrant osteoblast-like cell is the cell of origin for osteosarcoma [34]. Osteosarcomas express high levels of RANK, RANKL, and OPG, implying a global dysregulation in signalling between osteoblasts and osteoclasts [35]. Signalling pathways involved in normal bone development, such as Hedgehog (Hh), Notch, and WNT, have also been implicated in osteosarcoma pathogenesis [21]. Expression of *NOTCH* genes has been correlated with metastatic phenotypes [36], and expression of *GLI2*, a transcriptional target of Hh, has been associated with worse survival in a cohort of 51 osteosarcoma patients [37]. In addition, WNT signalling is more active in osteosarcomas [38,39], which could be linked to its role in angiogenesis in addition to that in bone formation [40]. Cross-talk between these pathways has been suggested to be important for therapy resistance [21,36,38,41], which will be discussed in further detail in a following section.

### 3.2. The Tumour Microenvironment

The interaction between malignant tumour cells and their microenvironment contributes importantly to tumour propagation, invasion and, subsequently, metastasis [42]. In particular, an inflammatory microenvironment can promote malignant features and metastatic potential [23], and the microenvironment can modulate the efficacy of antitumour therapies [42]. The bone microenvironment is closely coupled to the immune system [43], making the latter of particular importance in understanding the microenvironment of bone-derived tumours.

Tumour cells can trigger immune cells by expression of tumour-specific antigens [44]. However, tumours can also suppress immune activation by expressing factors that downregulate antitumoral immunity. One down-regulator of immunity is cytotoxic T-lymphocyte antigen 4 (CTLA-4), which subdues T-cell activation early on in the immune response [45]. Maintenance of peripheral immune tolerance at later stages of immunity is gained by the expression of programmed death ligand-1 (*PD-L1*) [46]. PD-L1 (or PD-L2 and B7-H3) on tumour cells binds to its receptor, programmed death-1 (PD-1), on immune cells to yield apoptosis, anergy, and tolerance of the T cells [47,48]. Upregulation of inhibitory receptors on T cells can also occur in cancer (a hallmark of T-cell exhaustion) due to persistent high antigenic loads from tumour cells [49], which, together with expression of inhibitory factors on tumour cells, allows for large downregulation of immune responses. Highly immunogenic tumours such as melanoma are ineffectively cleared due in part to their ability to express *PD-L1* and *CTLA-4* [50,51]. Given the close association between bone tissue and the immune system, it has been hypothesized that osteosarcoma may use similar mechanisms to escape immune recognition. In support of this, one study showed that osteosarcoma primary patient samples with positive PD-L1 immunohistochemistry had worse event-free survival as compared to patients with PD-L1-negative osteosarcoma [47]. The PD-L1-positive samples also exhibited a higher level of immune cell infiltration than their PD-L1-negative counterparts [47]. This is in line with the finding that osteosarcomas with high levels of immune infiltration are enriched in immune downregulation pathways, including PD-1 signalling and the CTLA-4 pathway [52]. Collectively, this suggests that osteosarcoma may utilize PD-L1, PD-L2, B7-H3, and CTLA-4 to counteract immune recognition.

Monocyte-derived macrophages, a diverse type of immune cell, are central regulators in bone biology, as they act as osteoclast precursors in the presence of M-CSF and RANKL [53]. Macrophages found in the tumour-surrounding area are known as tumour-associated macrophages (TAMs) and are involved in regulating local immunity, angiogenesis, and tumour cell migration [54,55]. TAMs consist of various subpopulations that are frequently being dichotomized into the M1 or M2 type of macrophage according to their differentiation and function, although there is evidence supporting the probability of a phenotypic continuum between those extremes [56]. The M1-polarized macrophage is considered to have antitumour functions, whereas the M2 subtype is an alternatively activated population that is believed to promote tumour formation and maintenance [57].

Osteosarcoma tumours exhibit widespread macrophage infiltration [58]. Expression of genes linked to tumour-associated macrophages has been found to be correlated with lower risk of metastasis, good response to chemotherapy, and better overall survival [59]. Although macrophage number is positively correlated to osteosarcoma survival, the presence of M2-polarized macrophages is associated with poor prognosis [59] and a shift in the M1/M2 balance to favour the M1 subtype correlated to nonmetastatic disease [60]. In a mouse osteosarcoma model, inhibiting M2 polarization of TAMs was found to prevent the formation of lung metastases [61]. These data suggest M2-TAMs as potential drivers in the metastatic potential of osteosarcoma. This may be due to a role for M2-TAMs in T-cell suppression, i.e., the number of M2-TAMs has been shown to correlate with the abundance of suppressive T-lymphocytes in osteosarcoma, and depletion of M2-TAMs leads to an increase in T-lymphocyte proliferation [62]. One study found that a high ratio of cytotoxic (CD8+) T-cells to regulatory (FOXP3+) T-cells was a positive prognostic factor for osteosarcoma patients [63], in line with evidence from dogs showing that a decrease in this ratio was associated with decreased survival [64]. In addition, one study reported that CD8+ cytotoxic T-cell tumour infiltration correlated with better survival [65]. Together these findings indicate that active immunity is positive for outcome in osteosarcoma patients. In line with this, osteosarcoma mouse models fare better after osteomyelitis [66], dogs with postoperative infections exhibit longer survival [67], and osteosarcoma patients with postoperative infections have slightly increased survival rates [68].

The RANK/RANKL/OPG system involved in bone remodelling is also involved in immune system regulation [25], i.e., survival of dendritic cells is promoted by RANKL, and RANKL drives dendritic cells to promote naïve T-lymphocyte proliferation and survival [69,70]. Given the overexpression of RANK, RANKL, and OPG observed in osteosarcoma [35], it may be that this perturbs the microenvironment. Indeed, many osteosarcomas exhibit T-cell infiltration, and the presence of upregulated immune factors is a negative prognostic factor [23]. High levels of the inflammatory cytokine IL-6 are detected in patients with Paget’s disease of bone [71], which is thought to be a driver in the pathogenesis of this disease [72]. The higher risk for osteosarcoma in this patient group [22] could then be a reflection of not only extensive bone remodelling but also the inflammatory bone microenvironment. One small study found that higher values in inflammatory prognostic scores, such as Glasgow Prognostic Score, C-reactive protein, neutrophil-lymphocyte ratio, and platelet-lymphocyte ratio, were negatively correlated with overall survival in osteosarcoma patients [73], suggesting that certain types of inflammation may be coupled to poor outcome. Understanding which characteristics of inflammation in osteosarcoma are promoting or repressing metastatic growth is crucial for development of immunomodulatory treatment strategies and for understanding how immunotherapy can influence conventional chemotherapy (see below).

### 3.3. Genomic Instability and Cell Cycle Regulation

Genomic instability and unchecked cell cycle progression are common hallmarks of cancer [74]. PARP1 (poly(ADP-ribose) polymerase 1) is a DNA repair molecule associated with malignancy [75]. Upon DNA damage, PARP1 is activated and detects DNA single-stranded breaks (SSBs) to aid in their resolution via base excision repair (BER), thus preventing their transition to double-stranded breaks (DSBs) [75]. If DSBs occur, spontaneously or due to therapy, PARP1 induces phosphorylation of H2AX (γH2AX) and recruitment of BRCA1/2, which aid in DSB repair and activation of DNA-damage checkpoints [75]. In this way, PARP1 promotes cell survival in the presence of DNA damage. PARP1, γH2AX, BRCA1, and BRCA2 are found in the nuclei of osteosarcoma tumours, and poor survival in osteosarcoma patients has been linked to expression of these factors [76]. Nevertheless, it was reported that up to 80% of osteosarcoma patient samples exhibit a “BRCAness,” indicating that they share similarities to tumours harbouring germline mutations in *BRCA1* or *BRCA2* making them deficient in DNA repair pathways [77]. In line with this, the osteosarcoma cell line LM7 was found to display BRCAness and was sensitive to a PARP inhibitor [78]. Expression of *ERCC1*, a member of the nucleotide excision repair pathway, has also been correlated with poor survival in osteosarcoma patients [79]. Together these data indicate that a central factor in osteosarcoma progression may be its ability to bypass DNA damage checkpoints. Furthermore, a defect in DNA damage responses can not only lead to chemotherapy resistance [80], but also confer synthetic lethality [81].

Consistent with this notion, nearly all osteosarcomas harbour mutations in the tumour suppressor gene *TP53*, and many have mutations in the cell cycle checkpoint regulator *RB1* [20,82,83,84]. Cells with mutations in *TP53* are impaired in their DNA damage response, and continue with mitosis despite catastrophic DNA alterations [20]. This contributes to widespread genomic instability, and loss of *TP53* is one of the main hallmarks of many cancers [85]. Given the global aberrations in *TP53* in osteosarcoma, it is of no surprise that these tumours are highly genomically unstable [82,86,87]. Furthermore, patients with germline mutations in either *TP53* or *RB1* run a 10–100-fold higher risk of developing osteosarcoma than the general population [84,88]. Additionally, patients with mutations in different members of the RecQ family of DNA helicases, which are important for DNA replication and repair, have a predisposition for osteosarcoma [89]. One case report describes two brothers, who both developed osteosarcoma, with germline mutations in *ATRX*, a gene encoding a transcriptional regulator with, among other roles, important mitotic functions [90]. These findings further implicate a role for malfunctioning DNA damage pathways and cell cycle checkpoints as key elements in the pathogenesis of osteosarcoma and have to be taken into account for rational therapy design.

### 3.4. Cellular Signalling Pathways

Receptor tyrosine or serine/threonine kinases are a family of cell-surface receptors that are central in key processes such as differentiation, proliferation, and cell cycle control in normal and cancerous cells [91]. The ErbB/HER family of receptors comprises Her1 (epidermal growth factor receptor (EGFR)), Her2, Her3, and Her4 [92]. Lung, breast, stomach, and colorectal cancers are examples of malignancies linked to mutation or overexpression of members of this family [92]. Perturbations in the ErbB/HER family have been found in osteosarcoma as well. Her2 protein and *HER2* gene overexpression has been detected in osteosarcoma tumours via immunohistochemistry and quantitative PCR [93,94]. This overexpression has been linked to poor patient outcomes [93,95] as well as improved survival [96]. The presence and role of Her2 has been debated, with several studies also reporting that Her2 is not expressed at all in osteosarcoma [97,98]. EGFR and Her4 have also been detected in osteosarcoma samples [94], and one study reported EGFR protein overexpression in 50% (6/12) of osteosarcoma cell lines tested [99]. High expression of EGFR has been associated with good clinical outcome [100] as well as metastasis and relapse in a small patient study [99]. Hence, while the exact nature of the role of ErbB proteins in osteosarcoma remains unclear, they may be useful targets in selective therapies (see below).

IGF-1R is another tyrosine receptor kinase that has been linked to osteosarcoma pathogenesis, and enhanced tumorigenesis in human and canine cell lines [101]. Tumour neovascularization relies on factors such as vascular endothelial growth factor (VEGF) and its tyrosine kinase receptor VEGFR [102]. Expression of VEGF has been associated with osteosarcoma metastasis and poor prognosis [103,104]. High expression of platelet-derived growth factor (PDGF)-AA, an activator of VEGF, and its receptor (PDGFR) has been suggested to be a biomarker for progressive osteosarcoma [105].

Protein kinases, frequent downstream targets of receptor kinases, are responsible for intracellular protein phosphorylation and are vital for proper intracellular signal transduction, and disruptions have been linked to several malignancies including osteosarcoma [106]. Mitogen-activated protein kinases (MAPK) have also been found to be highly active in osteosarcoma cell lines [107]. The mammalian target of rapamycin (mTOR) is a serine/threonine protein kinase belonging to the phosphoinositide-3-kinase (PI3K)-related kinase family. It is a central controller in many important cellular functions, and many cancers, including osteosarcoma, have an abnormally high activity of mTOR [108]. Enhanced activity of these pathways favours rapid cell growth, with the hope that the advent of targeted therapies against these protein kinases will aid in the treatment of highly proliferative tumours such as osteosarcoma (see below).

More recently, dysregulation of the tumour-suppressive Hippo pathway has been implicated in osteosarcoma [109]. The stem cell transcription factor Sox2, a supposed marker for cancer stem cells and highly expressed in osteosarcoma [110], downregulates the Hippo activators merlin (Nf2) and WW domain-containing protein 1 (WWC1) and upregulates the Hippo suppressor Yes-associated protein 1 (YAP) [109].

Following binding to its receptors, transforming growth factor-β (TGF-β) signals mainly through Smad3/4 proteins that act as transcription factors following nuclear translocation in both osteosarcoma cells and its microenvironment, promoting osteolysis, angiogenesis, and metastases (for review, see [111]). Inhibiting TGF-β signalling in murine osteosarcoma models led to reduced lung metastasization [112].

## 4. Osteosarcoma Therapies

The vast genomic instability, heterogenicity, and metastatic proficiency of osteosarcoma require a highly intensive combination of therapies. Although surgery (and sometimes radiotherapy [113]) is critical for local control of osteosarcoma, systemic control requires systemic treatment. Hence, the following sections focus on chemo-, immuno-, and targeted therapeutic strategies.

### 4.1. Chemotherapy

Conventional osteosarcoma treatment prior to (neoadjuvant) and following (adjuvant) surgery is based on systemic chemotherapy. Chemotherapeutic agents interfere with crucial cellular mechanisms, frequently leading to cell cycle arrest and apoptosis [114]. Importantly, this results in not only the death of cancerous cells but also damage to healthy cells. It has long been thought that the ultimate success of chemotherapy in osteosarcoma can be predicted by the grade of necrosis exhibited by the tumour following neoadjuvant therapy. Tumours exhibiting >90% or >95% necrosis are considered good histological responders, and those with less necrosis are considered poor histological responders [1,115]. Good histological response is associated with better event-free and overall survival [115,116]. Histological response is believed to be mainly influenced by the number, combination, types and doses of chemotherapy agents administered [117].

Cytotoxic therapies used in osteosarcoma include alkylating agents, antimetabolites, topoisomerase inhibitors, anthracyclines, or microtubule inhibitors. Alkylating agents such as ifosfamide and cisplatin work in a cell cycle-independent manner in which alkylated DNA adducts lead to DNA damage and cell death [114]. Methotrexate and gemcitabine are examples of antimetabolites, which are mainly S-phase-specific and hinder DNA replication directly or by interfering with dNTP synthesis [114]. Etoposide is a type of topoisomerase inhibitor which blocks DNA topoisomerase II (Top2), leading to DNA strand breaks and subsequently perturbations in transcription, replication, and mitosis [118]. Anthracyclines like doxorubicin inhibit DNA replication primarily via generation of DNA intercalations and inhibition of Top2 [114], while microtubule inhibitors work by distorting the mitotic spindle, resulting in mitotic arrest and apoptosis [119]. Although these classes of drugs can be used as monotherapy, they are most effective when used in combination. The goal of combination chemotherapy is to intensify chemotherapy exposure by circumventing dose-limiting toxicity of single drugs and avoid the issue of resistance to a single chemotherapeutic drug, thereby ultimately increasing antitumour efficacy. It is thus fundamental to combine drugs that are active as single agents and have different mechanisms of action, resistance, and toxicity [120]. Advantages of this strategy include the potential to avoid treatment resistance, emergence of additive or synergistic effects, and reduction of drug dose with subsequent reduction of toxicity [120,121]. Combination chemotherapy is undoubtedly pivotal for osteosarcoma treatment.

A combination of high-dose **m**ethotrexate with leucovorin rescue (HDMTX), doxorubicin (**A**driamycin), and cisplatin (**p**latin), so-called MAP, is the backbone of both neoadjuvant and adjuvant chemotherapy at centres in the United States and most of the Europe [122,123,124]. These three agents, along with ifosfamide, exhibit single-agent efficacy as well, though they can have higher effect when used in combination (reviewed in [125]). Addition of ifosfamide (I) to the MAP backbone (MAPI) in first-line therapy has not been proven to improve outcome in patients with nonmetastatic disease [116]. Similarly, an Italian study found that neoadjuvant and adjuvant MAPI did not improve histological response or survival and was related to more severe treatment-related toxicities when compared to MAP alone, though it did have a slight benefit as adjuvant therapy in patients with poor histological response to MAP [126]. These findings are in line with results from similar studies [127,128]. A small study by Kang et al. attempted to improve the histological response in osteosarcoma patients by using neoadjuvant MAP or compressed MAPI to align with the MAP schedule, as MAPI generally requires extra weeks of therapy prior to surgery. In the MAPI group, 71% of patients had a good histological response versus 42% in the MAP group, though this difference, possibly due to the small cohort, was not significant [117]. Despite a trend towards better histological response, the MAPI group did not have improved survival than MAP [117]. Thus, MAPI is only recommended for use in patients with poor histological response to MAP in some countries. The European and American Osteosarcoma Study (EURAMOS) group attempted to improve outcome in poor histological responders following traditional MAP therapy by adding high-dose ifosfamide and etoposide to adjuvant MAP (MAPIE) in the EURAMOS-1 trial. Unfortunately, this did not result in survival benefit in these patients and gave an additional burden of toxicity [129]. These results indicate that MAP is an effective combination therapy. Furthermore, while predictive for outcome, the prognostic value of histological response following neoadjuvant treatment does not confer a viable stratification strategy for intensified chemotherapy with drugs generally active against osteosarcoma. As a consequence of these and other clinical trials, EURAMOS-1 concluded not to recommend tailoring adjuvant chemotherapy based on histological response [124,129,130].

Though MAP is the standard treatment in most countries, other drug combinations are used as well. For instance, doxorubicin-carboplatin-ifosfamide (API-AI) is a feasible alternative first-line therapy, particularly suitable for patients who would not tolerate HDMTX or cisplatin-based regimens [131]. In France, standard chemotherapy consists of methotrexate and etoposide-ifosfamide (M-EI) for children and adolescents and API-AI for adults [132]. The M-EI protocol is based on the randomized OS94 paediatric trial, which demonstrated that children and adolescent osteosarcoma patients receiving M-EI had a similar outcome as those receiving methotrexate combined with doxorubicin [128]. The adult API-AI protocol is based on a phase II trial showing that the API-AI group had similar response rates as a group receiving API-AI plus HDMTX [133], with similar findings reported in a phase III trial [134]. A recent French study reported that the M-EI and API-AI protocols yielded similar survival outcomes in 18- to 25-year-old patients, though histological response was better in the M-EI group (60%) than the API-AI group (41%) [132]. These findings indicate that osteosarcoma chemotherapy can be effective using combinations excluding either doxorubicin, methotrexate, or cisplatin. It would of course be interesting to speculate that biomarker-guided choice of the therapy regimen might yield better outcomes assuming subgroups exist that are particularly susceptible to either chemotherapeutic drug.

Second-line chemotherapy regimens for patients with relapsing disease are not as well-defined as first-line therapies [135]. A retrospective analysis of event-free survival in patients with refractory/recurrent osteosarcoma involved in seven different phase II trials demonstrated insufficient activity of all therapies tested [136], indicating inherent difficulties in treating this patient group. Therapies that have shown some success in improving outcome for patients with relapsing disease include carboplatin and etoposide [137] and cyclophosphamide and etoposide [138], and the search for other therapies that improve survival in the patient group is ongoing. Cyclophosphamide might be particularly interesting for patients that have already received ifosfamide due to clinical evidence that those two drugs, despite their similar modes of action, might not be cross-resistant [139]. The DNA-binding agent trabectedin leads to DNA damage and apoptosis and is approved for the treatment of soft tissue sarcoma [140]. A phase II trial of trabectedin in relapsing osteosarcoma patients did not improve survival, however, though the authors suggested it may be useful in combination with doxorubicin and cisplatin due to its ability to suppress MDR1 expression by inhibition of the orphan nuclear receptor steroid and xenobiotic receptor (SXR) [141,142,143] (see resistance section). This has, to our knowledge, yet to be tested. Interestingly, two patients homozygous for the wild-type Asp1104 small-nucleotide polymorphism of the DNA repair-associated ERCC5 gene were sensitive to trabectedin, suggesting that this drug could be of use in patients with aberrant DNA repair genes [144]. Another group of agents that has shown some effect in the treatment of recurrent osteosarcoma are microtubule inhibitors.

Microtubule-destabilizing agents such as vincristine are effective in the treatment of solid and haematological tumours, and stabilizing agents such as the taxanes, docetaxel and paclitaxel, are commonly used in breast and lung cancer chemotherapy [119]. Osteosarcoma trials from the 1970s and 1980s demonstrated clinical effects of vincristine in combination with methotrexate in adjuvant osteosarcoma therapy [145,146,147], though the ultimate conclusion of these trials was that vincristine in itself was not efficacious and that the results of these trials were due to methotrexate alone. Initial testing by the Pediatric Preclinical Testing Program (PPTP) demonstrated response in some osteosarcoma xenografts treated with paclitaxel [148], cabazitaxel, and docetaxel [149]. Docetaxel in combination with gemcitabine is used as a second-line treatment for some patients with recurrent osteosarcoma, though with contradictory results, with objective responses reporting to range from 0% to 46% ([150] and references within). Eribulin is a novel microtubule inhibitor that has been approved for treatment of some malignancies [151]. The PPTP demonstrated that out of the six osteosarcoma xenografts tested, four were sensitive to the drug (three with a complete response, one with stable disease) and two were resistant (progressive disease) [152]. These results are supported by in vitro data showing that eribulin is cytotoxic in osteosarcoma cell lines due to cell death as a result of microtubule instability and cell cycle arrest [153]. These promising results led to a phase II trial of eribulin in patients with refractory or recurrent osteosarcoma. The findings of this trial were disappointing, however, with 0 of the 19 patients reaching the 4-month time point without progression [154]. As of yet, no microtubule inhibitors are incorporated in current first-line osteosarcoma therapy.

Approximately, 10% of osteosarcoma patients harbour macrometastases at diagnosis [155]. Primary metastatic osteosarcoma is treated according to the same principles as nonmetastatic disease: neoadjuvant MAP-based chemotherapy, surgery when possible, and adjuvant MAP-based chemotherapy [155]. Ten-year survival in this patient group is approximately 20%, with the number of metastases and grade of incomplete surgical resection being directly correlated with poorer outcome [155]. Patients with recurrent osteosarcoma with or without metastases have survival rates similar to metastatic osteosarcoma. Predicting the likelihood of a patient to develop metastatic disease would be beneficial in choosing more successful therapies and a large systemic literature review attempted to pinpoint factors linked with recurrent metastatic disease. This proved to be complicated, however, due to the lack of large-scale studies evaluating prognostic factors, and no definite conclusions could be made [156]. Nonetheless, a consistent trend in metastasis-free survival was seen in patients with good histological response, indicating a predictive value of this parameter. Other factors found to be associated with increased risk for metastasis risk were tumour localization (in particular, axial tumours when radical surgery was not achieved), tumour size/volume, histological subtype (telangiectatic versus chondroblastic), and age [156,157].

Given that combination chemotherapy is able to delay and reduce the occurrence of overt lung metastases, it is tempting to speculate that a prolonged low-intensive maintenance chemotherapy might be able to improve survival in osteosarcoma patients. This regimen is fundamental for long-term outcomes in acute lymphoblastic leukaemia [158] and has been recently demonstrated to improve outcome in high-risk rhabdomyosarcoma [159].

### 4.2. Adverse Effects of Chemotherapy

The implementation of neoadjuvant and adjuvant chemotherapy has greatly increased survival rates of osteosarcoma, but this comes at a cost. Due to their rather unspecific mode of action, systemic chemotherapeutic agents target both cancerous and healthy cells, often leading to toxic side effects as a result of damage to healthy tissues. These side effects can range from moderate to life-threatening ailments, which can cease once treatment is stopped or lead to permanent damage that affects the quality of life in long-term survivors of childhood cancer after treatment. Almost all chemotherapy patients experience transient myelosuppression and gastrointestinal symptoms, which can usually be managed by supportive care [114]. Nonetheless, bone marrow suppression and mucositis can be so severe that patients require hospitalization or are forced to interrupt or discontinue therapy altogether, which can reduce the received cumulative doses of chemotherapy and might increase the likelihood for therapy resistance to develop, reducing chances of survival [160].

Ototoxicity can be a permanent consequence of cisplatin therapy, with its incidence being as high as 78% in osteosarcoma survivors and being correlated to cumulative dose and age [161,162]. Sodium thiosulphate (STS) can inactivate cisplatin, but preclinical evidence shows that a delayed application of STS does not reduce cisplatin antitumour efficacy [163]. In localized hepatoblastoma, addition of sodium thiosulphate was shown to reduce ototoxicity without compromising survival [164]. However, in a study recruiting a more heterogenous cancer population, post hoc analyses revealed a worse outcome for patients with metastatic disease [163,165]. Hence, there is a risk that addition of STS to osteosarcoma treatment might jeopardise its oncological efficacy. Localized application of STS, however, might be a viable option to protect from ototoxicity (and systemic side effects of STS) [166]. Other serious and permanent chemotherapy-related adverse effects observed in osteosarcoma patients are nephrotoxicity and cardiotoxicity, both of which can be fatal.

Methotrexate and its metabolites can precipitate in renal tubules, leading to acute renal insufficiency and failure. Because it is primarily eliminated by the kidneys, this precipitation increases renal exposure to methotrexate, thus increasing its nephrotoxicity in addition to its other side effects [167]. Leucovorin rescue is administered in order to minimize cellular HDMTX toxicity, though this has no effect on nephrotoxicity. Instead, aggressive hydration and alkalinization of urine, which discourages methotrexate precipitation, are routine in preventing HDMTX-related kidney damage. Despite these measures, an estimated 1.8% of osteosarcoma patients have nephrotoxicity associated with HDMTX, and fatal renal failure still occurs in some patients. Standard treatment for methotrexate-induced acute kidney injury is application of glucarpidase, which inactivates both methotrexate and 7-hydroxymethotrexate and can frequently prevent the need of haemodialysis [167,168]. Adults, in particular, have been reported to be more prone than children and adolescents to the detrimental side effects of HDMTX [169], and the need for HDMTX in this group has been debated [170], with some studies reporting similar survival rates from HDMTX-excluding protocols [132,171].

Cisplatin and ifosfamide can also be nephrotoxic, though their effects are generally less acute than those induced by HDMTX. Cisplatin-related renal dysfunction presents as decreased glomerular function and electrolyte imbalances (Fanconi syndrome) due to its damaging effects on tubular epithelial cells. Glomerular filtration (GFR) is reduced in 60% of children receiving median cisplatin doses of 500–600 mg/m^2^ (standard doses for osteosarcoma), though this effect is mild and commonly improves once treatment is stopped [172]. Hypomagnesemia as a result of cisplatin therapy is reported in up to 90% of patients, which persists in 12–20% of cases [173]. Ifosfamide also affects the GFR and can induce tubular damage, which can lead to tubular dysfunction resulting in chronic kidney damage and tubulopathy a decade after treatment in 10% and 30% of patients, respectively [174]. Though its effects are often mild, ifosfamide can cause fatal renal damage at high cumulative doses such as those seen in relapsing osteosarcoma patients [175]. Patients receiving combinations of cisplatin and ifosfamide have not been reported to experience clinically relevant severe renal toxicities [176]. One severe side effect of ifosfamide is bladder toxicity, as its metabolite acrolein can cause haemorrhagic cystitis. This side effect can effectively be prevented by increasing fluid intake and administering sulfhydryl donors (in particular Mesna) that react with acrolein to form a non-toxic product [114].

Doxorubicin is the main culprit for cardiotoxicity that occurs in some osteosarcoma patients. It is associated with acute/subacute cardiomyopathy within 1 year of treatment, late cardiomyopathy in decades following treatment, and cardiac-related death in osteosarcoma patients and survivors [177]. Cardiotoxicity is most likely underestimated in paediatric populations since it has been suggested that it may take more than 20 years before clinical symptoms arise [178]. Doxorubicin’s cardiotoxicity is dose dependent, with significant occurrence after cumulative doses of 300–450 mg/m^2^ and a systolic cardiac dysfunction in more than a quarter of patients 15 years after treatment [179]. Young patients (18–25 years of age) in a French study receiving a cumulative dose of 420 mg/m^2^ doxorubicin had a 39% incidence of acute cardiotoxicity, compared with 23% of patients of those who received 137.5 mg/m^2^ doxorubicin cumulatively [132], illustrating this dose dependence. The conventional cumulative dose of doxorubicin in standard MAP-based osteosarcoma therapy is 450 mg/m^2^ [116], and patients receiving lower cumulative dose or dose intensity have poorer event-free survival [180,181]. Doxorubicin doses exceeding 550 mg/m^2^ are associated with a significantly increased risk of heart failure [167], which is of particular concern in patients with relapsing disease who need additional cycles of chemotherapy. Symptomatic cardiomyopathy, observed as clinical presentation of congestive heart failure, has been reported to range from 1.5% [182] to 2.2% [183] in osteosarcoma survivors. In the 1.7% of osteosarcoma survivors with symptomatic cardiomyopathy observed in one study, almost 50% experienced cardiac-related death (6/13) and almost 50% of the survivors required heart transplant (3/7 alive) [184], underlining that congestive heart failure is of great concern in osteosarcoma survivors.

Although doxorubicin has been found to alter other cellular pathways, it primarily exerts its cytotoxic effects via DNA intercalation and Top2 inhibition. Humans express two Top2 isoenzymes: Top2α, which is most prevalent in highly proliferating malignant and non-malignant cells, and Top2β, which is more abundant in quiescent cells such as cardiomyocytes [185]. Doxorubicin intercalates with DNA and binds to Top2α to form Top2-doxorubicin-DNA complexes, resulting in the inability to repair DNA breaks [185]. In contrast, inhibition of Top2β in cardiomyocytes is believed to be responsible for cardiomyopathy [185,186]. Knocking out Top2β in mice protects them from doxorubicin-induced cardiomyopathy [187], suggesting that depleting Top2β concurrently with doxorubicin treatment may be cardioprotective.

Dexrazoxane is a Top2 inhibitor and iron chelator. It exerts a cardioprotective function in the presence of doxorubicin that has been linked to its ability to block Top2β [188,189]. Dexrazoxane has been found to reduce doxorubicin-associated cardiotoxicity without affecting treatment efficacy in women with breast cancer [190] and in children with acute lymphoblastic leukaemia [191]. One study found that osteosarcoma patients receiving dexrazoxane were protected from the cardiotoxic effects of high cumulative doses of doxorubicin (600 mg/m^2^) without impaired tumour response or increased risk for secondary neoplasms [177]. Another study in paediatric osteosarcoma patients receiving 375–600 mg/m^2^ doxorubicin following dexrazoxane administration reported similar results: prevention of cardiac dysfunction and heart failure without reduction in treatment efficacy [192]. Cardiac changes were not entirely prevented, however, with alterations seen in left ventricular anatomy especially in girls [192]. Nonetheless, these findings demonstrate that dexrazoxane is able to protect from doxorubicin-induced cardiotoxicity in osteosarcoma patients, and suggest that cumulative doxorubicin doses may be increased with its use, which might advocate clinical trials with further escalation of cumulative doxorubicin doses beyond 600 mg/m^2^.

### 4.3. Immunotherapy

The advent of immunotherapy has been ground-breaking in the treatment of selected cancers [193]. Immunotherapy as a strategy against osteosarcoma was introduced already in 1970 at the Karolinska Hospital. In the initial nonrandomized trial, researchers administered human interferon-α (IFNα) as adjuvant therapy to all patients presenting with nonmetastatic disease between 1971 and 1985 [194]. They found that the interferon group fared similarly to the group receiving standard MAP therapy, though they concluded that the results were difficult to interpret due to clinicopathological differences between the patient groups [194]. The promising results prompted the authors to add IFNα in the treatment of patients with unresectable, nonmetastatic osteosarcoma [195]. These patients exhibited a 63% progression-free survival after 5 years [195], leading to the conclusion that further, randomized trials needed to be launched in order to fully understand the effect of interferon in the treatment of osteosarcoma [196].

The large randomized EURAMOS-1 study attempted to improve outcomes for good histological responders by incorporating recombinant interferon α-2b (IFN-α-2b) to adjuvant treatment. IFN-α-2b modulates the immune system to yield antiproliferative and proapoptotic effects in several cancer models including osteosarcoma, making it a promising therapy [197]. IFN-α-2b in combination with MAP for good histological responders did not, however, improve outcome in these patients [130]. In the Scandinavian study (see above), a pool of leukocyte-derived interferons was used, while only a specific recombinant interferon was used in the EURAMOS trial, raising the hypothesis that it may be more effective to use a mixture of interferons in osteosarcoma treatment. Another immunomodulating cytokine tested in osteosarcoma is granulocyte-macrophage colony stimulating factor (GM-CSF), which has shown some promising results in the treatment of Ewing sarcoma and melanoma [198]. GM-CSF had no impact on lung metastases or outcome in osteosarcoma patients, however [199]. More promising results have been achieved with other immune system modulators.

Liposomal muramyl tripeptide phosphatidylethanolamine (L-MTP-PE) is a synthetic analogue of an immunogenic mycobacterial cell wall component [200]. L-MTP-PE is thought to be tumoricidal via release of inflammatory cytokines and pro-inflammatory molecules following activation of macrophages, specifically the M2 population, and monocytes [200]. A phase III clinical trial reported improved overall survival when L-MTP-PE was combined with systemic adjuvant osteosarcoma therapy for patients with nonmetastatic disease [116], and another study showed a trend towards improved survival in patients with metastatic disease with this same regime [201]. Due to the multifactorial study design, the possibility of an interaction between L-MTP-PE and ifosfamide could not be excluded, however, leading many to question the actual effect of L-MTP-PE on outcome in osteosarcoma patients [202]. As a result, L-MTP-PE has yet to be FDA-approved and is authorized for use in only a handful of European countries [203]. Nonetheless, these results suggest that the right kind of immune activation might have an important role to play in improvement of osteosarcoma survival.

The advent of monoclonal antibodies against immune checkpoints has been a breakthrough for cancer immunotherapy. Survival in osteosarcoma has been found to directly correlate with the level of infiltration of CD8+ T-cells in patient samples [65], suggesting that targeting immune checkpoints may be a fruitful strategy against osteosarcoma. A phase I trial of the anti-CTLA-4 antibody ipilimumab in heavily pretreated paediatric patients with relapsed or refractory disease yielded stable disease in two of the eight osteosarcoma patients involved in the trial, suggesting moderate activity [204]. One out of three adult osteosarcoma patients with metastatic disease treated with the anti-PD-1 antibody nivolumab demonstrated partial response, with one other having stable disease [205]. A phase II trial of the anti-PD-1 antibody pembrolizumab in adult osteosarcoma patients resulted in 1 patient with partial response and 6 with stable disease out of the 22 patients who completed the trial [206]. In a murine model of advanced osteosarcoma, immune checkpoint inhibition with three antibodies—anti-Tim-3, anti-PD-L1, and anti-OX-86—in combination with tumour debulking surgery resulted in better overall survival and no lung metastases in survivors [207]. These results illustrate that immune checkpoint therapy is active in some osteosarcoma patients, but most likely requires a combination with other treatment modalities.

Targeting CSF-1 receptor (CSF-1R) signalling is a recent immunotherapy strategy that can alleviate the immunosuppressive properties of myeloid-derived suppressor cells (MDSCs) [208]. Tumour-promoting M2-TAMs are a subtype of MDSCs and are dependent on CSF-1R signalling [209]. Blocking CSF-1R leads to a repolarization of TAMs towards the antitumour M1 subtype [210]. Higher numbers of MDSCs are associated with metastasis in canine osteosarcoma [211]. In human osteosarcoma, M2-TAMs are linked to metastasis and poor survival, whereas M1 polarization correlates with nonmetastatic disease and better survival [59,60]. Osteosarcoma cell lines with overexpression of CSF-1R ligands CSF-1 and IL-34 have a propensity to polarize TAMs to the M2 subtype, resulting in more progressive disease in a murine osteosarcoma model [212]. Blocking CSF-1R in a mouse osteosarcoma model led to fewer lung metastases and prolonged survival [213]. This strategy also yielded results in canine osteosarcoma—the CSF-1R inhibitor toceranib showed efficacy in 11 out of 23 patients treated [214]. Addition of a CSF-1R inhibitor potentiates and overcomes resistance to checkpoint inhibitors in other solid tumour models [215,216], suggesting that combination of a checkpoint inhibitor with a CSF-1R inhibitor could have potential in osteosarcoma therapy.

Some studies have reported effects of anti-inflammatory agents on osteosarcoma. One case report describes remission in a patient with metastatic, MAP-resistant osteosarcoma after treatment with adjuvant anti-inflammatory therapy alone [217]. This patient initially presented with nonmetastatic osteosarcoma and underwent neoadjuvant MAP according to protocol. After 3 cycles, however, lung metastases were discovered, and the patient refused adjuvant salvage therapy. Instead, he was treated with celecoxib, a cyclo-oxygenase 2 (COX2) inhibitor, and thalidomide, which is thought to inhibit tumour angiogenesis [218], perhaps via a COX2-dependent mechanism [217], though some suggest that it may promote tumour degradation via an angiogenesis-independent COX2-related mechanism [219]. One month after beginning treatment, the lung metastases cleared up and the patient could walk with less pain. This suggests that anti-inflammatory factors are also important in the regulation of osteosarcoma. COX2 was found to be expressed in 73% of childhood osteosarcomas, suggesting it to be a therapeutic target [220]. In addition, the MG-63 osteosarcoma cell line was found to be sensitive to parecoxib, another COX-2 inhibitor [221]. Our laboratory has found that celecoxib and thalidomide inhibit proliferation in a dose-dependent manner in several osteosarcoma cell lines (unpublished results), in line with published results for celecoxib showing dose-dependent death in cell lines MG-63, U2-OS, and MNNG/HOS [222].

Glucocorticoids are essential components in the treatment of lymphoblastic malignancies [223] and are also widely used in the treatment of inflammatory disorders for their dampening of inflammatory responses [224]. Treatment of osteosarcoma cell lines with dexamethasone, a synthetic glucocorticoid, has been shown to lead to modified gene expression, inhibition of cell proliferation, and promotion of apoptosis [225]. Our laboratory has found that dexamethasone inhibits growth in a subset of osteosarcoma cell lines at high doses (unpublished results).

Collectively, these findings indicate that both the promotion and suppression of inflammation can be exploited for the treatment of osteosarcoma. Osteosarcoma is a complex disease, and inflammation itself is a complicated and multifaceted process. Promoting “good” inflammation that encourages the immune system to recognise and eliminate tumour cells, such as with MTP-PE, may be most beneficial in patients with progressive disease, as these tumours have most likely already used escape mechanisms to avoid detection [42]. On the other hand, high levels of inflammation are present at early disease stages [42], suggesting a role for anti-inflammatory therapy. More research is required to determine the timing and type of immunomodulatory treatments.

### 4.4. Targeted Therapies

The development of targeted therapies for a subset of cancer-related receptors and their ligands has led to effective treatment of some solid malignant tumours [226]. In particular, the use of the monoclonal Her2-antibodies trastuzumab and/or pertuzumab in combination with docetaxel has significantly improved outcomes in patients with Her2-positive breast cancer [227,228]. Given the high level of Her2 protein expression reported in some osteosarcoma tumours and cell lines [93,95], Her2-antibodies are of interest in the treatment of osteosarcoma as well. Attempts at adding trastuzumab to MAP therapy were disappointing, however, with patients showing neither reduced development of metastatic disease nor improvement in overall survival [229]. This is in line with other studies that question the influence of Her2 on osteosarcoma development [230]. Nonetheless, some propose that antibody-based therapies against Her2 may be beneficial in a subset of osteosarcoma patients that exhibit high *HER2* expression in the primary tumour [231]. EGFR (Her1) has also been suggested to play a role in the pathogenesis of osteosarcoma [94,99,100]. Although inhibiting EGFR with gefitinib has been successful in the clinical treatment in a subset of non-small cell lung cancer [232], it has not been effective in inhibiting proliferation or survival in osteosarcoma cell lines on its own [233]. However, treating osteosarcoma cell lines with gefitinib improved the effect of methotrexate and doxorubicin, leading the authors to conclude that EGFR may contribute to chemotherapy resistance [234]. There is additional preclinical evidence that active EGFR signalling is important for osteosarcoma progression [235]. The small molecule VEGFR and EGFR inhibitor ZD6474 was able to reduce tumour growth in an osteosarcoma mouse model and had a synergistic effect with the COX2-inhibitor celecoxib in vitro and in vivo [222]. These findings demonstrate that these targeted therapies may be useful in osteosarcoma when combined with other agents.

A better approach for targeted therapy in osteosarcoma may be to target a whole family rather than an individual member. Afatinib is an irreversible inhibitor of the ErbB family that blocks all signalling from dimers formed by all family members that has been effective in the treatment of advanced squamous cell lung cancer [236,237]. Cruz-Ramos et al. recently reported that afatinib inhibited proliferation in osteosarcoma cell lines HOS, Saos-2, SJSA-1, U2-OS, and MNNG and was able to reduce cell migration and invasion in a metastatic (MNNG) and nonmetastatic (HOS) cell line [236]. In HOS cells, afatinib treatment correlated with reduced protein phosphorylation of Her2/EGFR and their downstream signalling molecules Akt and Erk1/2, suggesting that inhibition of ErbB-dependent pathways underlies the cellular effects [236]. Thus, afatinib may warrant clinical evaluation against osteosarcoma.

Sorafenib is an unspecific tyrosine kinase inhibitor with activity against MAPK, BRAF, VEGFRs, and PDGFR [238] that has been effective in the treatment of renal, hepatic, and thyroid cancers [239]. Given its broad activity against various tyrosine kinases, and the heterogenous nature of kinase aberrations in osteosarcoma [21,107,108,240], it has been hypothesized to be useful in osteosarcoma therapy. In preclinical osteosarcoma models, sorafenib was able to restrict cell proliferation and reduce metastasis formation [107], which led to a phase II clinical trial in which patients with relapsed, unresectable osteosarcoma following standard MAP therapy were administered sorafenib [241]. Initial results were promising, with a 46% progression-free survival after 4 months, a 16% increase from the primary endpoint of 30% [241]. In an attempt to increase sorafenib’s activity, the authors concurrently supressed mTOR, which had been shown to inhibit tumour growth in osteosarcoma mouse models [242]. A subsequent phase II clinical trial was launched where patients with relapsed, unresectable osteosarcoma following standard MAP therapy were given both sorafenib and the mTOR inhibitor everolimus. The findings of this study were disappointing, however, with patients achieving a 6-month progression-free survival of 45%, short of the primary endpoint of 50% [243]. This demonstrates the difficulties in applying “targeted” therapies in an unselected patient cohort.

Regorafenib is a multikinase inhibitor similar to sorafenib but with broader activity [244]. A phase II randomized study sought to measure the effects of this agent in patients with advanced or metastatic osteosarcoma that had failed to respond to standard therapy regimes. This study found that 65% of patients in the regorafenib group had no disease progression after 8 weeks compared with 0% of those in the placebo group and that the treatment group had a progression-free survival of 3.8 months versus 1 month in the placebo [245]. These results were supported by a similarly designed phase II study, which reported that patients receiving regorafenib had significantly improved progression-free survival, i.e., 3.6 months compared to 1.7 months for those receiving placebo [246]. These promising findings suggest that regorafenib may be useful in the treatment of progressive or metastatic osteosarcoma and warrant further studies into the effects of this agent in combination with conventional chemotherapy. Cabozantinib is a VEGFR2 tyrosine kinase inhibitor that is also able to inhibit MET, the protein product of the TPR-MET transforming oncogene derived from an osteosarcoma cell line [247]. A phase II trial by the French Sarcoma Group found that cabozantinib treatment in patients with advanced osteosarcoma yielded 12% (5/42) partial response and 33% (14/42) 6-month progression-free survival, suggesting that it too may be promising in future treatment protocols [248].

Bisphosphonates are another class of agents with promising preclinical activity in osteosarcoma [249]. Through inhibition of farnesyl pyrophosphate synthase, bisphosphonates interfere with prenylation of small Rho GTPases, ultimately inhibiting osteoclast activity. Additional mechanisms like antiangiogenic properties are thought to contribute to their activity [250]. Increased activity of RhoA has recently been identified to result from fusion proteins that involve Rab22a. Those fusions were found in 2/37 metastatic osteosarcoma patients and were associated with lung metastases in mouse models [251]. Zoledronate is a nitrogen-containing bisphosphonate that has been shown to be effective in hindering osteosarcoma tumour progression and metastasis in mouse models (reviewed in [252]). Given these results, the OS2006 trial attempted to improve outcome in osteosarcoma patients by combining zoledronate with chemotherapy and surgery in a phase III clinical trial. The results were disappointing, however, with poorer event-free 3-year survival observed in the zoledronate group compared with the control group (57.1% versus 63.4%) [253]. This suggests that bisphosphonates, despite promising preclinical evidence, may not be of benefit in osteosarcoma patients, and further studies are needed to investigate the reason for this discordance. 

IGF-1R is a tyrosine kinase receptor also involved in bone pathways. A monoclonal antibody against IGF-1R (R1507) was able to restrict growth in osteosarcoma xenograft tumours alone and in the presence of the mTOR inhibitor rapamycin [254], but did not have a clinical effect in a phase II clinical trial [255]. A bispecific antibody targeting IGF-1R and EGFR was able to inhibit tumour growth and prevent lung metastases in a mouse osteosarcoma model [256], suggesting that it may have potential as a therapy for metastatic osteosarcoma, though this has yet to be tested clinically. RANK activation is another pathway thought to be central to osteosarcoma progression. Inhibiting RANKL with siRNA did not hinder osteosarcoma progression in a mouse model, however [257].

Inhibiting PARP1 has been successful in the treatment of malignancies that are deficient in DNA-repair pathways, especially those with “BRCAness” [258]. Consistent with this, one study showed that osteosarcoma cell lines with BRCAness (MG-63, ZK-58, Saos-2, and MNNG-HOS) were sensitive to the PARP1 inhibitor talazoparib, while a cell line with a heterozygous BRCA2 mutation (U2-OS) was not [259]. Talazoparib was also synergistic with the chemotherapy agent temozolomide (TMZ) in the induction of apoptosis in the MG-63 and ZK-58 cell lines [259]. These findings are supported by a similar study that found that administration of the PARP1 inhibitor olaparib induced cell death in osteosarcoma cell lines U2-OS, Saos-2, MG-63, and KHOS/NP [76]. The possibility of using PARP inhibition to treat resistant osteosarcoma is currently underway (see Section 5 for details).

Glycoprotein nonmetastatic B (gpNMB) is a type I transmembrane glycoprotein expressed in normal cells involved in tissue repair, adhesion, and growth, and its overexpression has been found in several cancers including osteosarcoma [260]. Roth et al. showed that gpNMB was expressed in 62/67 tested osteosarcoma patient samples, all primary samples tested had gpNMB mRNA overexpression, and that gpNMB was expressed on the surface of all 19 tested cell lines, suggesting that it may be readily targeted [261]. Glembatumumab vedotin (GV) is an antibody–drug conjugate consisting of a gpNMB-specific monoclonal antibody coupled to the microtubule inhibitor monomethyl auristatin E (MMAE) [260] that has shown effect in the treatment of gpNMB-expressing breast cancer [262]. GV inhibits tumour growth in osteosarcoma xenograft models [263] and cell lines [261], indicating that it may be effective in patients. A phase II trial for GV in patients with refractory or relapsed osteosarcoma showed that of the 22 patients enrolled, only 1 patient had partial response and 2 had stable disease and reported that there was no correlation between gpNMB expression and GV response [264], indicating that GV did not have substantial clinical efficacy.

Together these data demonstrate that while progress has been made in targeting specific molecular mechanisms underlying osteosarcoma in in vitro models, this has yet to translate to clinical outcomes. One reason for this may be that targeted therapies require careful patient selection, and should, therefore, be implemented based on the presence or absence of specific biomarkers. Though some attempts have been made to individualize therapy based on tumour characteristics, more research is needed in this area [265]. One possibility is that targeted therapies have a wider range of use when combined with traditional chemotherapy in order to promote therapy sensitivity and combat resistance.

A summary of therapeutic targets that have led to clinical trials in osteosarcoma is presented in Table 1.

## 5. Mechanisms Underlying Therapy Resistance

Osteosarcoma is frequently referred to as a drug-resistant tumour [9], and even patients with good response to primary therapy usually require very high doses of a combination of chemotherapy agents, with the most effective agents (MAPI) achieving cure rates of only a 30–40% when administered alone [267,268,269,270]. For doxorubicin, there is clear evidence that the cumulative administered dose correlates with antitumour efficacy [180]. Two different studies furthermore inferred, by comparison of differentially treated adults and children [271] and comparison of outcomes from different treatment protocols [272], that higher doses of MTX yield superior outcomes. Those findings were corroborated by pharmacokinetic studies that determined a beneficial cut-off of a plasma peak concentration of more than 1000 µM and a minimum number of six HDMTX courses [273]. Similarly, reduction of per-protocol chemotherapy intensity due to toxic complications has a negative impact on survival outcomes [274]. However, several randomized trials did not see improved survival through therapy intensification with the addition of two or more chemotherapeutic agents as compared to the standard arm [129,275,276]. Thus, at least 30% of osteosarcoma patients cannot be cured by surgery and current chemotherapy combinations. In other words, systemic chemotherapy is frequently unable to eradicate residual osteosarcoma cells. Since combination chemotherapy is effective for the majority of patients, this is indeed suggestive of the presence or emergence of chemotherapy-resistant osteosarcoma cells. Although targeted therapies and immunotherapy might contribute to future therapy improvements, elucidating mechanisms of resistance would allow the development of strategies to target them during ongoing therapy in order to sensitise osteosarcoma to standard combination chemotherapy.

Cancer cells can achieve therapy resistance by many different mechanisms depending on the agent and cellular target. Chemotherapy resistance in osteosarcoma can be linked to perturbations in mechanisms underlying drug build-up in the cell, intracellular detoxification, apoptosis, DNA damage repair, signal transduction, tumour microenvironment (tumour stem cells), and immunity. In addition, mutations in the drug target can confer resistance. Oftentimes, these mechanisms render tumour cells not only resistant to a specific drug but also to seemingly unrelated drugs. This is called multidrug resistance (MDR) [277].

MicroRNAs (miRNAs) are short regulatory RNAs that negatively modulate protein expression at a post-transcriptional and/or translational level, and studies have revealed that certain miRNAs can be used as therapeutic targets in cancer, especially in combating drug resistance [278]. In osteosarcoma, many miRNAs have been discovered that are linked to chemotherapy resistance that have been hypothesized to be targets for novel molecular therapies. The role of miRNAs in the evolution and potential treatment of drug-resistance in osteosarcoma has been thoroughly covered in a recent review [279] and will not be discussed in its entirety here, though key miRNAs linked to specific pathways will be mentioned.

Mechanisms triggering multidrug resistance in osteosarcoma are demonstrated in Figure 2 and will be discussed in the following sections, highlighting how these pathways can be targeted in order to overcome resistance where applicable.

### 5.1. Limitations of Drug Delivery

Because many solid tumours have a poorly formed vascular system, delivery of chemotherapeutic agents is often inefficient. In addition, drugs must penetrate several layers of tissue for optimal effect, making it difficult for solid tumours to be effectively treated by intravenously administered chemotherapy agents [280]. These limitations to drug delivery have been proposed to be a main cause of methotrexate resistance in osteosarcoma [281] (Figure 2A). To overcome this, chemotherapeutic agents can be coupled to nanocarriers that can increase drug delivery at tumour sites. At the same time, this technology can protect the drug from rapid clearance and prolong drug circulating time, making nanocarriers attractive in overcoming therapy resistance in osteosarcoma. Different types of nanocarriers are currently under investigation for use in osteosarcoma therapy at the preclinical and clinical level, though none have been approved to date. Wang and colleagues have recently published an extensive review on nanocarriers in osteosarcoma drug delivery that we recommend for details [282]. An acidic tumour microenvironment can furthermore reduce activity of cytotoxic drugs [283].

### 5.2. Decreased Intracellular Drug Accumulation and Target Specificity

Crucial to the function of anticancer agents is their ability to accumulate within or in proximity to malignant cells in order to exert their actions. Thus, a cell that efficiently accumulates a drug and continuously keeps it in the vicinity of its target is more sensitive to the given therapy. Tumour cells often present with mechanisms that minimize drug influx, increase drug efflux, and change the drug target (Figure 2B–D).

#### 5.2.1. Decreased Cellular Influx

Impaired transport of methotrexate is a common mechanism of resistance in osteosarcoma cells (Figure 2B). Methotrexate is an inhibitor of dihydrofolate reductase (DHFR) that provides tetrahydrofolate, which is a one-carbon donor for de novo purine and thymidine biosynthesis, thus depleting the dNTP pool required for DNA replication and repair. It primarily enters cells via the reduced folate carrier (RFC encoded by *SLC19A1*) at the cell membrane with folate receptor alpha only playing a minor role [284]. Decreased expression of RFC has been shown to mediate methotrexate resistance in osteosarcoma [285,286,287]. Overall, 65% of osteosarcoma biopsy samples were found to have decreased RFC expression, with low expression more commonly found in patients with poor histological response to chemotherapy [285]. Another study demonstrated that RFC expression was somewhat decreased in chemotherapy-insensitive osteosarcoma samples and that decreased expression of RFC was more common in primary tumour samples than in metastases [288]. This may suggest that RFC expression is an intrinsic, rather than acquired, resistance mechanism. In line with this, another study found that RFC protein levels were lower in primary versus recurrent tumour specimens, with significantly lower RFC levels also correlating with poor histological response [289]. Yet this same study also found that high RFC protein levels correlated with osteosarcoma recurrence.

RFC with the single-point mutation Leu291Pro renders the carrier unable to translocate the substrate across the cell membrane, which confers methotrexate resistance in osteosarcoma cell lines. Some degree of methotrexate resistance is achieved by reducing transport rates as seen in cell lines harbouring any one of the mutations Ser4Pro, Ser46Asn, and Gly259Trp [290]. Studies into the RFC gene demonstrate that gene deletion does not underlie reduced RFC protein expression, as no differences in gene copy number between parental osteosarcoma cell lines and their methotrexate-resistant variants have been found [291]. Tumour samples with high frequencies of sequence alterations in the RFC gene have been coupled to poor histological response, though it was not clear if these alterations were germ-line or tumour specific as normal tissue was not analysed [292]. In addition, RFC methylation status and polymorphisms modulate its expression [293]. Further studies are warranted to elucidate the role of alterations in the RFC gene and/or protein in methotrexate resistance in patients.

One way to bypass RFC-mediated methotrexate resistance is by using a methotrexate-like drug that does not require RFC for transport. The antifolate trimetrexate—due to its lipophilicity—does not require RFC for cellular influx and has been tested in a phase II study in patients with relapsed or refractory osteosarcoma. Altogether, 13% (5/38) patients receiving trimetrexate had a response [294]. A phase I trial investigated the role of trimetrexate together with HDMTX in patients with recurrent osteosarcoma, though no results of this study have been published till date (ClinicalTrials.gov Identifier: NCT00119301). Alternatively, the antifolate pralatrexate with a higher affinity for RFC might compensate for lower RFC expression [295]. This compound has undergone clinical trials in haematopoietic malignancies and has been approved for the treatment of patients with refractory or recurrent peripheral T cell lymphoma [296], making it of possible interest in osteosarcoma. Interestingly, the effects of MTX in osteosarcoma cell lines can be largely inversed in vitro by supplementation of deoxynucleosides important for the DNA salvage pathway. This suggests that osteosarcoma might be particularly vulnerable to nucleoside analogues [297].

#### 5.2.2. Increased Cellular Efflux via ATP-Binding Transporters

Overexpression of members of the ATP-binding cassette (ABC) family of efflux transporters is a common mechanism of multidrug resistance in cancer cells [298] (Figure 2C). This family includes P-glycoprotein (P-gp), multidrug resistance-associated protein (MRP1 ABCC1 and MRP2 ABCC2), and breast resistance-associated protein (BCRP or ABCG2), all of which have been found to contribute to multidrug resistance in osteosarcoma [277]. The membrane-bound pump P-gp promotes nonspecific removal of cytotoxic drugs and is encoded by the gene multidrug resistance 1 (MDR1, also known as ABCB1) and has been found to be strongly associated with drug resistance in paediatric solid tumours [299]. Although *MDR1* gene expression has not been found to correlate with outcome in osteosarcoma [300], overexpression of *MDR1* has been found in doxorubicin-resistant cell lines [301], and altered ABC transport pathways are strongly linked to doxorubicin resistance. Many studies have reported that expression of P-glycoprotein is a strong clinical prognostic factor. Patients with osteosarcoma exhibiting positive immunohistochemistry for P-glycoprotein have poorer relapse-free and overall survival than their negative counterparts [302,303]. In one study, over 50% (27/53) of P-gp-positive patients experienced relapse and had a 47% cumulative survival probability compared with rates of 17% and 83%, respectively, in patients negative for P-gp [303]. P-glycoprotein expression levels are not associated with differences in clinicopathological tumour features or histological response, however [302,303]. In patients intending to receive doxorubicin-containing chemotherapy, P-glycoprotein expression at diagnosis was found to be a significant prognostic factor, with overexpression being coupled to adverse outcomes [304]. On the other hand, some studies have failed to couple P-glycoprotein expression with survival outcome in osteosarcoma [305,306,307]. These discrepancies could be due to sample size, differences in methods, and non-standardization of analysis protocols. One of these studies did link P-gp to metastasis formation, however, as 72% (13/18) of patients with high P-gp developed metastases [306]. Interestingly, this study also showed that good responders had less P-gp staining than poor responders and that pretreated osteosarcoma tumours had a tendency to have increased P-gp staining compared with untreated tumours, though neither finding reached statistical significance [306]. Another study found that 68% of lung metastases were P-gp positive compared with 32% of primary tumour samples, suggesting an expression gain in P-gp in metastases [308]. Although its role as a prognostic factor for survival remains unclear, it seems that P-gp has a clinical predictive value in terms of risk for metastasis and that its expression may be induced following treatment initiation, though its expression has not been correlated to histological response. High expression of the neurotrophic growth factor pleiotrophin, which upregulates P-gp, has been found to be linked to poor overall and disease-free survival in a retrospective study of 133 osteosarcoma patient samples [309]. It has also been found that polymorphisms in *ABCC2* correlate with response to chemotherapy [310].

In vitro studies have also demonstrated the importance of ABC transporters in resistant osteosarcoma and the potential to overcome resistance by using them as drug targets. A study by Yang et al. found that osteosarcoma cell lines U-2OS and Saos-2 selected for resistance to paclitaxel had higher P-gp levels and demonstrated cross resistance to other P-gp substrates such as doxorubicin, docetaxel, and vincristine. However, cells continuously exposed to paclitaxel at doses that confer resistance were able to maintain paclitaxel sensitivity when it was given in the presence of NSC23925, a small molecule inhibitor of P-gp [311]. NSC2395 was not able to rescue already resistant cells, however, suggesting that it may be more beneficial to prevent rather than treat drug resistance. In a similar study, the alkaloid anti-inflammatory compound tetrandrine was found to prevent paclitaxel-induced MDR in the osteosarcoma cell line U2-OS. In cells treated with paclitaxel alone, the promoter activities of MDR1 and nuclear factor (NF)-κB, as well NF-κB binding to the *MDR1* promoter, were enhanced. Upon treatment with tetrandrine together with paclitaxel, the expression and activity of NF-κB were significantly decreased, thus preventing P-gp overexpression [312]. Interestingly, NF-κB is also highly involved in the expression of many proinflammatory factors [313], so it may be that reducing inflammation in combination with drugs known to cause MDR may prevent resistance. Glucocorticoids prevent inflammation by inhibiting the NF-κB pathway [314], and our laboratory is currently investigating the role of the glucocorticoid dexamethasone in drug resistance and synergy in osteosarcoma cell lines (unpublished results).

Wu and colleagues reported that upregulation of P-gp by pleiotrophin promotes doxorubicin resistance while knockdown of pleiotrophin enhances chemosensitivity in osteosarcoma cell lines [309]. Knock-out of *ABCB1* in multidrug resistant osteosarcoma cell lines KHOSR2 and U-2OSR2 using CRISPR-Cas9 gene editing technology was able to restore doxorubicin sensitivity but had no effect on cisplatin sensitivity [315]. This is in line with the finding that cisplatin resistance is most often a result of altered DNA repair mechanisms and not cellular depletion of the drug ([316] and see following section). Targeting ABCB1/ABCC1 with the inhibitor CBT-1 (Tetrandrine) in chemotherapy-resistant osteosarcoma cell lines was able to restore sensitivity to doxorubicin as well as second-line therapy drugs etoposide, Taxotere, and vinorelbine, making this an interesting option in the treatment of refractory or recurrent osteosarcoma [317]. As mentioned above, it has been found that trabectidin inhibits transcriptional activation of MDR1 [318] and is therefore potentially useful in the treatment of resistant osteosarcoma. Thus, targeting or altering ABC transporter expression in combination with chemotherapy may be a useful in preventing resistance and treating resistant osteosarcoma.

### 5.3. Alterations to Drug Targets

The cytotoxic effects of doxorubicin are mostly based upon its intercalation with DNA and binding to Top2α (encoded by *TOP2A*) in rapidly proliferating cells to form Top2-doxorubicin-DNA complexes, which ultimately leads to DNA strand breaks. In senescent cells, it can bind Top2β (encoded by *TOP2B*), which can lead to apoptosis in cells such as cardiomyocytes [185]. Alterations in Top2 can thus prevent the action of doxorubicin and cause chemoresistance (Figure 2D). One study found that rearrangements in *TOP2* genes were found in 40–67% of tumours from a paediatric population. Both amplification and deletion of *TOP2A* in patient samples was associated with good histological response, though poorer overall and event-free survival was also correlated with its amplification. Tumours with amplified *TOP2B* had better event-free survival, whereas *TOP2B* deletion conferred poorer event-free survival [319]. These results suggest that *TOP2* gene status may be a prognostic factor in survival and response to chemotherapy, at least in a paediatric population. In line with this notion, lower levels of Top2β mRNA were found in a doxorubicin-resistant cell line compared with its parental cell line [301]. Concurrent administration of doxorubicin with dexrazoxane, a Top2β inhibitor that protects against the cardiotoxic effects of doxorubicin, does not affect treatment efficacy [192], however, suggesting that alterations in Top2β alone does not underly doxorubicin resistance.

Methotrexate and its polyglutamates are competitive inhibitors of the enzyme dihydrofolate reductase (DHFR), and methotrexate polyglutamates inhibit two enzymes in the purine synthesis pathway that require folate coenzymes. Inhibition of DHFR impairs regeneration of tetrahydrofolate from dihydrofolate, resulting in a tetrahydrofolate deficit in replicating cells. This inhibits purine and thymidine synthesis and subsequent DNA replication, ultimately causing apoptosis. Mutations in DHFR leading to reduced affinity for methotrexate are important in the acquisition of methotrexate resistance for some cancers, including acute leukaemia [320]. High levels of DHFR expression have been described in methotrexate-resistant osteosarcoma cell lines, and xenografts with high DHFR expression exhibit methotrexate resistance [287]. Gene duplications in DHFR are common mechanisms underlying methotrexate resistance in patients with acute lymphoblastic leukaemia, and these duplications are common in patients with concurrent p53 mutations [321]. Only 10% of osteosarcoma patient samples in one study were found to have increased DHFR expression at biopsy and none have evidence of gene amplification, however [285]. Nonetheless, one study found that DHFR expression was lower in initial osteosarcoma biopsy specimens than in metastases in a paediatric population [288]. Lack of functional Rb may contribute to increased activity of DHFR in methotrexate-resistant osteosarcoma tumours. One study in soft tissue sarcomas reported that cell lines with dysfunctional Rb have a two-to-four-fold higher DHFR expression than those with normal Rb, which was coupled to methotrexate resistance [322]. In osteosarcoma, associations have been made between E2F transcription factors, which influence Rb control of gene expression, and DHFR mRNA expression in cell lines [323]. This implies that dysfunctional DHFR in osteosarcoma may be due to Rb signalling aberrations rather than gene duplication. Inconsistent with this idea is the finding by Serra and colleagues that methotrexate resistance in cell lines with intact Rb signalling (methotrexate-resistant U2-OS) was associated with increased levels of DHFR, whereas the methotrexate-resistant variant of the Rb-negative Saos-2 cell line did not exhibit any signs of DHFR abnormality [291]. Using comparative genomic hybridization, two studies by the same group identified amplification of DHFR in methotrexate-resistant variants of the U2-OS cell line [324,325]. One study found that miR-215 led to reduced expression of DHFR, but that miR-215 overexpression actually increased methotrexate resistance in osteosarcoma cell lines. This discrepancy was explained by the fact that miR-215 overexpression led to a p53-dependent growth inhibition and that low-proliferating tumours are more resistant to S-phase specific cytotoxic drugs, such as methotrexate [326]. Thus, alterations in DHFR may be linked to p53 status in osteosarcoma tumours as well, and targeting p53 may be a way to overcome methotrexate resistance due to p53-mediated DHFR alterations.

### 5.4. Intracellular Drug Modifications

It is crucial that a drug remains active following its cellular accumulation so that it can function on its target. Cells possess pathways that allow them to detoxify drugs once they have entered. Glutathione-S-transferases (GSTs) are a family of Phase II detoxification enzymes that catalyse the conjugation of glutathione (GSH) to a variety of compounds that can enter a cell, resulting in their inactivation (Figure 2E). The human cytosolic GSTP1 has been linked to inactivation of many agents including anticancer drugs [327], and some have found that GSTP1 overexpression confers chemoresistance in various cancer types [328]. In canines with osteosarcoma, higher GSTP1 expression was associated with significantly shorter median remission and survival times [329]. One study from 60 human osteosarcoma samples found that overexpression of GSTP1 protein via immunohistochemistry in surgical samples following neoadjuvant therapy was correlated with poor histological response, but this overexpression was not observed in pretreatment biopsy samples [330]. This suggests that increased GSTP1 protein expression may be induced during therapy, making cells resistant to therapy.

In line with this, cisplatin resistance has been associated with increased levels and enzymatic activity of GSTP1 in osteosarcoma cell lines [331]. Another study found that GSTP1 expression was induced in osteosarcoma cell lines following treatment with doxorubicin or cisplatin. Overexpression of GSTP1 in the osteosarcoma cell line Saos-2 led to increased resistance to doxorubicin and cisplatin, and GSTP1 suppression in the HOS cell line caused more apoptosis and DNA damage in response to these drugs [332]. A significant correlation was also found between higher GSTP1 transcript levels and low growth inhibition following doxorubicin treatment in osteosarcoma xenografts [333], and poor histological response has been found to be increased in germ-line variants of GSTP1 [310]. Another investigation into the genotype variations of GSTP1 found that individuals with GSTP1 Val/Val had a shorter survival than those with the IIe/IIe genotype [334], though a different study found that patients with Val actually had a significantly better response to chemotherapy [335]. The gene polymorphism GSTP1 rs1695 GG genotype and G allele has been more often found in osteosarcoma patients with poor response to chemotherapy, poor event-free survival, and poor overall survival [336]. Taken together, these findings support a role for GSTP1 in chemotherapy resistance in osteosarcoma and open up the possibility that targeting GSTP1 could be a useful therapeutic strategy especially in overcoming cisplatin resistance. The anticancer compound NBDHEX inhibits GSTP1 and has been found to have synergistic effects with cisplatin when co-administered in cisplatin-resistant cell lines [331], making it an interesting candidate that warrants further studies.

Methotrexate is polyglutamated upon entry into the cell, and both methotrexate and its polyglutamated products are competitive inhibitors of DHFR to lead to replication defects and apoptosis. Methotrexate polyglutamates are retained better in cells than methotrexate, so cells that can effectively accumulate methotrexate polyglutamates are more sensitive to methotrexate, while those that can hydrolyse the polyglutamates to shorter chains are more resistant [320] (Figure 2E). In line with this, overactivity of the enzyme γ-glutamyl hydrolase yields a ~70% reduction in accumulation of the methotrexate polyglutamate 10-propargyl-5,8-dideazafolate, which confers methotrexate resistance in a hepatoma cell line [337]. It is also known that lack of methotrexate retention due to lack of polyglutamylation is a major mechanism underlying therapy resistance in acute lymphoblastic leukaemia [338]. Whether this mechanism is also involved in the methotrexate resistance seen in osteosarcoma is unknown, though it can be speculated that it plays a role. In patients with localised disease, high methotrexate serum concentration during neoadjuvant treatment was one of the most important predictive factors for good histological response [273,339]. To achieve this, osteosarcoma patients must be administered high doses of methotrexate, perhaps due to its reduced accumulation in cells at low doses due to intrinsic resistance mechanisms such as decreased polyglutamylation. In support of this, one study found that spindle and kinetochore associated complex subunit 1 (SKA1) overexpression was associated with de novo methotrexate resistance and poor 5-year survival in a cohort of patients. SKA1 overexpression led to a downregulation of folylpoly-γ-glutamate synthetase (FPGS), a key enzyme in the polyglutamylation of methotrexate, via Ska1 interaction with the RNA polymerase II subunit RPB3. In cell lines, overexpression of SKA1 also led to methotrexate resistance, while SKA1 downregulation was able to restore drug sensitivity [340]. These findings present a new target to which new agents can be directed in order to overcome methotrexate resistance due to alterations in its intracellular metabolism.

### 5.5. Inhibition of Apoptosis and Cell Cycle Regulation

Chemotherapy frequently causes catastrophic DNA damage, which consequently leads to cell death. Cell cycle arrest, on the other hand, delays apoptosis by allowing the cell to repair DNA damage before it eventually re-enters the cell cycle. Alterations in apoptosis or cell cycle signalling could, therefore, underly chemotherapy resistance in tumour cells (Figure 2F). Overexpression of the genes encoding prohibitin (an antiproliferative protein) and rhoA (involved in apoptosis) decreased drug sensitivity to approximately 52% and 59%, respectively, in the Saos-2 osteosarcoma cell line due to inhibition of apoptosis [341], illustrating the importance of apoptosis in osteosarcoma chemosensitivity. The following sections will focus on key apoptosis pathways found to be disturbed in resistant osteosarcoma.

#### 5.5.1. B Cell Lymphoma 2

B cell lymphoma 2 (Bcl-2) was found as the product of an oncogene, and it is the founding member of a family of proteins involved in cell death signalling. This family localizes to the mitochondrial outer membrane and encompasses antiapoptotic proteins including Bcl-2 and Bcl-xL as well as the proapoptotic proteins Bax, Bak, and Bad [342]. Osteosarcoma patients with high expression of Bcl-2 have lower long-term survival rates than those with low expression [343], and another study found that lung metastases had a higher frequency of positive Bcl-2 staining than primary tumour samples (84% versus 53%), though this difference was not significant most likely due to small sample size [308]. However, although two other studies also reported high Bcl-2 staining in osteosarcoma patient samples, neither were able to correlate Bcl-2 expression with survival [344,345]. One of these studies did find, however, that patients with a high Bax/Bcl-2 ratio had a poorer 4-year disease-free and overall survival [345], suggesting that dysregulation of apoptosis underlies treatment failure in some patients.

An in vitro study attempting to elucidate the role of apoptosis-related proteins in osteosarcoma found that inhibition of antiapoptotic Bcl-2 using lentivirus-mediated RNA interference increases doxorubicin sensitivity in a doxorubicin-resistant MG-63 cell line, which correlated with increased levels of apoptosis [346]. This suggests that promoting apoptosis by inhibiting antiapoptotic signalling could overcome chemotherapy resistance in osteosarcoma. Along these lines, promoting proapoptotic signalling yielded similar results. Upregulation of proapoptotic Bax as a consequence of bone morphogenetic protein (BMP) via runt-related transcription factor 2 (Runx2) was able to enhance apoptosis after etoposide treatment in the cell line Saos-2 [347]. These findings indicate that Bcl-2 inhibitors may be a feasible treatment for therapy-resistant osteosarcoma. Direct targeting of Bcl-2 family members is independent of p53 status since p53 lies upstream of the Bcl-2 pathway [348], which is especially relevant in osteosarcoma as most osteosarcoma tumours have p53 abnormalities [82]. Navitoclax is a Bcl-xL inhibitor that has been demonstrated to inhibit cell proliferation in two canine osteosarcoma cell lines [349]. Navitoclax has had promising response rates in clinical trials in chronic lymphocytic leukaemia (CLL), a cancer type highly dependent upon Bcl-2 pathways for survival, but with high levels of toxicity: severe thrombocytopenia in one-third of patients. This was attributed to targeting of Bcl-xL in platelets, prompting the search for a Bcl-2 specific molecule. Venetoclax is the result of reverse engineering of navitoclax and is a specific Bcl-2 inhibitor. Venetoclax has been shown to have good response rates in the treatment of relapsed or refractory CLL with manageable toxicities [350]. Given the activity of navitoclax in canine cell lines, it would be of interest to test venetoclax in human osteosarcoma cell lines, especially in combination with traditional therapies in resistant cells.

#### 5.5.2. TP53

The *TP53* gene is vital for proper cell cycle arrest and apoptosis upon DNA damage, and perturbations in *TP53*, including deletion and mutations, promote the malignant features of many cancers [85]. High levels of mutations in *TP53* are found in osteosarcoma [82]. The clinical importance of p53 and its contribution to therapy resistance in osteosarcoma is controversial.

In a study of 24 patient samples, loss of heterozygosity at the *TP53* locus was found in 54% of samples, with only 15% of patients in this group being sensitive to neoadjuvant chemotherapy, as measured histologically and radiologically, compared with 64% response in the 46% of patients without alterations in the *TP53* locus [351]. This suggests that *TP53* deletion is associated with chemoresistance. Consistent with reports for other malignancies, which showed that *TP53* mutations are correlated with increased p53 protein expression [352], one study reported that positive immunohistochemical staining for p53 in lung metastases conferred a 17% postrecurrence survival, compared with 64% survival in those who had lung metastases negative for p53 [308]. In line with this notion, all patients with the staining pattern bax(+)/bcl-2(-)/p53(+) on osteosarcoma tumour biopsy had relapsed after 4 years [345]. A meta-analysis including 499 patients found that p53-positive patients had a tendency to worse 2-year overall survival rates, but this finding was not significant and the study was unable to correlate p53 status with chemotherapy response. The authors concluded that response to chemotherapy is independent of p53 status but that gene alterations in *TP53* may nevertheless be associated with decreased survival [353]. In addition, one group reported lower constitutive levels of wild-type p53 protein in a cisplatin-resistant cell line (OST/R) compared with its sensitive parental cell line (OST) and that the OST/R cell line was unable to induce p53 following cisplatin exposure [354]. On the contrary, Tsuchiya and colleagues reported that transfection of wild-type p53 into p53-null Saos-2 cells increased cisplatin sensitivity [355]. Along these lines, overexpression of microRNA (miRNA) 140 (miR-140), which was associated with chemosensitivity in osteosarcoma tumour xenografts, was found to induce p53 expression and G1/G2 arrest in osteosarcoma cell lines with wild-type p53 (U2-OS) but less in cell lines with mutated p53 (MG-63) [356]. Contrarily, another study found that induction of p53 led to an 8-fold decrease in cisplatin sensitivity in Saos-2 cells under normal serum conditions (10%), while p53 induction during low serum conditions (1%) led to a 10-fold increase in cisplatin sensitivity [357], suggesting that p53′s role in chemotherapy resistance may vary dependent on extracellular conditions and soluble factors. Transfection of TP53-R273H, a p53 mutant found to be overexpressed in a drug- and apoptosis-resistant squamous cell carcinoma cell line, into p53-null Saos-2 osteosarcoma cells rendered cells resistant to doxorubicin and methotrexate and led to downregulation of apoptotic enzymes [358], suggesting that p53-dependent resistance to apoptosis is the cause for loss of chemosensitivity.

Both the nature of p53 aberrations in cancer and characteristics of osteosarcoma between patients are extremely heterogenic, perhaps rendering the search for a universal, unifying p53 alteration in osteosarcoma futile. The status of p53 may, however, be useful in the treatment of some individuals and should therefore not be disregarded entirely. One way to overcome p53-mediated drug resistance in candidate patients would be via its reactivation. The small molecule RITA (reactivating p53 and inducing tumour apoptosis) has been found to sensitize colon cancer cells to standard chemotherapy agents [359]. While this would not work for individuals with p53 deletions, it has the possibility to have an effect in patients with p53 mutations. Our laboratory is currently testing the efficacy of RITA in osteosarcoma cell lines, both as a mono- and combination therapy (unpublished results), and how its effects correlate with p53 status.

### 5.6. Alterations in DNA Repair Pathways

Three out of the four commonly used cytotoxic agents in osteosarcoma work by generating either direct (cisplatin and ifosfamide) or indirect (doxorubicin) DNA damage. Thus, enhancing DNA repair pathways is one mechanism how tumour cells gain resistance against DNA damage-inducing agents (Figure 2G). Cisplatin is the drug most extensively studied with regard to resistance linked to DNA repair, particularly because of evidence indicating that cisplatin resistance frequently leads to cross-resistance to other DNA-damaging agents used in osteosarcoma protocols [360,361]. Induction of alternate DNA repair pathways as a consequence of cisplatin treatment may explain why treatment intensification with ifosfamide and etoposide fail to improve survival [276].

The DNA repair pathway most often linked to chemotherapy resistance is nucleotide excision repair (NER), as it is the pathway primarily used to remove bulky DNA lesions such as those formed by chemotherapeutic agents [362]. The *ERCC* excision repair genes and corresponding proteins 1-5 (ERCC1-5) as well as xeroderma pigmentosum A (XPA) are involved in the NER pathways. It was found that pre-treatment osteosarcoma biopsy samples with positive ERCC1 immunohistochemistry had worse prognosis both with regard to event free and overall survivals, and that those being positive for both ERCC1 and ABCB1 had significantly worse prognosis [79]. On the other hand, low expression of *ERCC4* has been linked to poor histological response in a German study [363]. This suggests that NER is involved in chemotherapy resistance, perhaps together with P-gp, in these tumours. Because of this, attempts have been made to elucidate the predictive and prognostic value of genetic polymorphisms in *ERCC1* and *ERCC2* in chemotherapy response and survival. One study revealed a significant correlation between the polymorphism *ERCC2* A751C and poor response to cisplatin-containing therapy (45% response in those with at least one polymorphic allele versus 80% response in those with homozygous for common T allele) and shorter event-free survival (184 months compared with 240 months for individuals homozygous for common T allele) but did not find a correlation with survival and *ERCC1* variants [364]. One study linked *ERCC1* rs11615 CC alleles with a better clinical outcome [365], and another found a positive correlation between *ERCC1* C8092A genotypes and event-free survival, with patients carrying the C allele (CC and CA) having significantly longer event-free survival rates than those with the AAA genotype [366]. The CC genotype of *ERCC1* has been strongly correlated with survival in a previous study as well [367]. This strong correlation is thought to reflect a reduction in function in *ERCC1* associated with this polymorphism, which mediates defective NER and makes tumour cells sensitive to chemotherapy [366].

Another DNA repair pathway heavily involved in the repair of DNA damage cause by cytotoxic agents is base excision repair (BER), in which PARP1, PARP2, and Ape1 are important proteins. Ape1 has been associated with multidrug resistance and prognosis in many cancers [368]. One study revealed amplification of the *APE1* gene in 50% of tested cases and that high levels of Ape1 protein were expressed in 65% (37/57) of osteosarcoma samples. High Ape1 protein expression was found to be associated with local recurrence and/or metastasis [369]. Ape1 is also crucial in preventing hypoxia-related death and has been hypothesized to be involved in resistance to antiangiogenic therapy. In support of this, decreasing Ape1 expression levels with small interfering RNA (siRNA) was able to sensitize osteosarcoma xenografts to the antiangiogenic endostatin [370]. The miRNA miR-513a-5p has been found to supress *APE1* expression in osteosarcoma cell lines including HOS and U2-OS, rendering them radiosensitive [371]. In addition, miR-765 was found to sensitize osteosarcoma cells lines (HOS and 9901) and tumour xenografts to cisplatin due to downregulation of Ape1 [372], making targeting *APE1* through miRNA a treatment option for resistant osteosarcoma.

High PARP1 expression in osteosarcoma has also been correlated with shorter survival [76], suggesting a role for this BER protein in resistance as well. The cell lines U2-OS, Saos-2, MG-63, and KHOS/NP were sensitized to doxorubicin with concurrent treatment with the PARP1 inhibitor olaparib or when *PARP1* was knocked down with siRNA [76]. Olaparib and doxorubicin were found to have a synergistic effect in this study [76]. A phase II clinical trial is opening this year to test the effect of olaparib together with the ataxia telangiectasia and rad3 related (ATR) kinase inhibitor ceralasertib in patients with unresponsive or recurrent osteosarcoma (ClinicalTrials.gov Identifier: NCT04417062), and a phase II trial is underway to test the effect of olaparib alone in the treatment of refractory disease (ClinicalTrials.gov Identifier: NCT03233204).

Hence, targeting genes and proteins involved in the NER and BER pathway could provide an opportunity to circumvent resistance based on enhanced DNA repair mechanisms. A recent study by Fanelli and colleagues showed that silencing of *ERCC1, ERCC2, ERCC3, ERCC4*, or *XPA* was able to increase cisplatin sensitivity in the resistant U2-OS/CDDP300 and U2-OS/CDDP1 cells, and that silencing of *ERCC1, ERCC2*, or *ERCC4* increased cisplatin sensitivity in the parental cell line U2-OS [373]. The authors then screened known inhibitors of NER genes in order to find compounds that could increase cisplatin sensitization in vivo. NSC130813 and triptolide improved cisplatin efficacy in resistant and sensitive cell lines and displayed no evidence of cross-resistance with cisplatin [373]. These agents warrant further testing in a clinical setting.

### 5.7. Signal Transduction

There is also evidence inferring a role for perturbed signal transduction pathways in generation of chemotherapy resistance in osteosarcoma (Figure 2H). Gene expression analyses have revealed that chemotherapy-resistant osteosarcoma cell lines have a higher expression of several kinases compared with their non-resistant counterparts [373]. These kinases include *FGFR1, MAP2K3, MAPK1, MAPK3*, and *PIC3C3*. A screen of possible kinase inhibitors found that GDC0994 (targeting the MAPK pathway) and PD173074 (targeting *FGFR1*) were able to reduce the IC50 of cisplatin in resistant osteosarcoma cell lines [373]. Though further testing is needed, use of these compounds could be useful in sensitizing resistant osteosarcoma to standard treatment.

The receptor tyrosine kinases Her2 and VEGF have been thought to have potential as targets in the treatment of refractory or metastatic osteosarcoma. High Her2 expression has been correlated with poor histological response and poor outcome [93] and high expression of VEGF has been associated with worse disease-free and overall survival [374,375]. A clinical trial combining a Her2-inhibitor with conventional chemotherapy did not improve outcomes for metastatic patients [229], however, suggesting that targeting Her2 does not have a clinical effect on resistant disease, which is consistent with other findings correlating high Her2 levels with good clinical outcomes and those finding no correlation at all between Her2 and outcome in osteosarcoma [96,97,98]. The VEGFR inhibitor AZD2171 was found to have antitumour activity in osteosarcoma xenografts by the Pediatric Preclinical Testing Program [376]. The tyrosine kinase inhibitor sorafenib targets kinases including VEGFs and has been tested together with the mTOR inhibitor everolimus in patients with relapsed osteosarcoma, though this salvage therapy was not found to be superior [243]. Taking into account genomic studies highlighting the PI3K/mTOR pathway as a common vulnerability in osteosarcomas [377,378], it may be more successful to target these pathways in combination with neoadjuvant and/or adjuvant chemotherapy in order to prevent resistance rather than trying to manage overt resistance at later stages. In line with this, one study found that EGFR inhibition in osteosarcoma cell lines, while having only limited cytotoxicity on its own, was able to enhance the antiproliferative and antimigration effects of doxorubicin and methotrexate, suggesting that EGFR targeting may be one way to potentiate treatment with these drugs (see Section 3).

Another molecule with tyrosine kinase activity, IGF-1R, has been found to be highly expressed in osteosarcoma [379], and this high expression has been correlated with metastatic disease and poor overall survival [380]. One study found that blocking IGF-1R by the compound tyrphostin (AG1024) together with doxorubicin enhanced growth inhibition in osteosarcoma cell lines more than either compound alone. In addition, this study found that AG1024 had synergistic effects with doxorubicin even in the doxorubicin-resistant variant of the 143B cell line and increased doxorubicin sensitivity in this cell line [381], suggesting that inhibiting IGF-1R may be useful in treating resistant osteosarcoma. Robatumumab, a fully human neutralizing anti-IGF-1R antibody, was tested in a phase II study of patients with relapsed osteosarcoma or Ewing sarcoma. In the osteosarcoma groups, 3 of 31 patients with resectable metastases had a complete or partial response, while 0 of the 29 patients in the group with unresectable metastases showed response. Patients in the Ewing sarcoma group did not show favourable responses either, leading to the conclusion that this therapy was not useful in the treatment of relapsed disease in these patients [266]. Taking into account preclinical data, it may be of interest to use IGF-1R inhibition in combination with chemotherapy drugs in patients rather than as a monotherapy.

### 5.8. Autophagy

Autophagy is a process that degrades cellular organelles and proteins and maintains cellular biosynthesis. It can allow tumour cells to survive cellular stress by clearing damaged organelles and proteins, but can, at the same time, promote programmed cellular death under certain conditions. Tumour cells often exploit autophagy-related pathways to promote chemotherapy resistance and survival [382]. The chromatin-binding nuclear protein HMGB1 plays a role in facilitating autophagy following administration of cytotoxic agents in order to promote tumour cell survival [383]. In one study, doxorubicin, cisplatin, and methotrexate were found to upregulate expression of HMGB1 mRNA in osteosarcoma cell lines MG-63, Saos-2, and U2-OS. Subsequent knockdown of HMGB1 by RNA interference was able to restore chemotherapy sensitivity in the cell lines MG-63 and Saos-2, which correlated with increased levels of apoptosis. These results were confirmed in vivo using murine tumour xenografts. Overexpression of HMGB1 in cell lines allowed them to resist apoptosis when treated with doxorubicin, cisplatin, and methotrexate [384].

Another study demonstrated that chemotherapy resistance in osteosarcoma cell lines is dependent on both HMGB1 and autophagy [385]. The same was found for the cancer and autophagy-related gene *HMGN5* in osteosarcoma cell lines: overexpression led to chemotherapy resistance due to upregulation of autophagy, whereas knockdown led to sensitivity due to downregulation of autophagy [386]. The *HSP90AA1* gene was also found to be responsible for autophagy-dependent drug resistance in osteosarcoma [387]. Together, these results imply that autophagy allows osteosarcoma cells to resist apoptosis and that autophagy-promoting factors are induced by chemotherapy agents. This is an important resistance mechanism that can be used as a target in the development of therapies to combat drug resistance.

### 5.9. Cancer Stem Cells and Tumour Microenvironment

It has been suggested that cancer initiation, propagation, metastasis, and recurrence is driven by a small subpopulation of tumour cells termed cancer stem cells (CSCs). These cells can be identified by cell surface markers that are unique for the specific cancer type, which have been hypothesized to have implications in their targeting by novel therapeutic approaches. Like normal stem cells, CSCs can self-renew while producing differentiated daughter cells, and recent evidence has implicated the quiescent nature of CSCs in mechanisms of multidrug resistance. The surrounding microenvironment in which CSCs reside highly resembles a traditional stem cell niche and, similar to a niche, has been found to affect the ability of CSCs to grow, self-renew, resist drugs, invade, and metastasize. Targeting CSCs and their microenvironment could combat therapy resistance and aid in the prevention and/or resolution of cancer recurrence or metastasis (Figure 2I) [388].

The exact role of CSCs in therapy resistance in osteosarcoma has yet to be defined, though several theories exist. One study found that the CSC cell line 3 AB-OS were strongly positive for CD133, a marker for pluripotent stem cells, and expressed higher levels of the ABC transporter gene ABCG2, which correlated with a high drug efflux capacity, and high expression of anti-apoptosis genes [389]. Another study linked a population of osteosarcoma CSCs derived from the MNNG/HOS cell line to high expression of drug efflux transporters P-glycoprotein and BCRP [390], further implicating a role for targeting transport mechanisms in the treatment of resistant osteosarcoma. The MG-63 osteosarcoma cell line was found to possess an ability to form spherical, clonal expanding colonies, called sarcospheres, in anchorage-independent, starved conditions. MG-63 sarcospheres resembled CSCs in their ability to self-renew and expression of stem cell-related genes and DNA repair enzymes. Resistance to doxorubicin and cisplatin was observed in MG6-3 sarcospheres, and administration of a DNA repair inhibitor, caffeine, was able to sensitize cells to these drugs. These findings suggest that osteosarcoma cell lines have subpopulations with the potential to form CSCs that are resistant to chemotherapy agents, perhaps via induction of extra DNA repair mechanisms [391]. A subsequent study correlated increased mRNA levels of aldehyde dehydrogenase 1 (ALDH1), which may contribute to enhanced drug detoxification, to drug-resistant MG-63 sarcospheres [392].

Another targetable element that could be used in treating resistance due to stem cell properties in osteosarcoma is S-phase kinase-associated protein 2 (Skp2). Skp2 has been found to positively regulate cancer stem cell populations and has self-renewal ability [393]. Overexpression of this protein in osteosarcoma cells was found to promote the epithelial-mesenchymal transition (EMT) observed in methotrexate-resistant variants of the cell lines U2-OS and MG-63, which allowed them to acquire enhanced invasive, migratory, and attachment abilities. Targeted shRNA against *SKP2* enhanced drug sensitivity in these cell lines. [394]. The novel Skp2 inhibitor, compound 25, was shown to have antitumour activities and to cooperate with chemotherapeutic agents to suppress cancer cell survival [393], making it of potential interest in osteosarcoma. Interestingly, nitidine chloride, an extract from the traditional Chinese medicine *Zanthoxylum nitidum*, was found to suppress the proliferative, migratory, and invasive ability of U2-OS cells due to repressed expression of EMT markers [395]. This study did not investigate whether nitidine chloride was involved in resistance, but it would be of interest to test this given its ability to suppress EMT that has been shown to be important in resistance in osteosarcoma.

Together, these studies imply that osteosarcoma CSCs can be targeted by selective inhibition of drug efflux systems, impairment of enhanced DNA repair pathways, downregulation of drug detoxification, promotion of pro-apoptotic factors, and repression of anti-apoptotic factors. Interestingly, these characteristics of CSCs are not unlike the characteristics described to be inherent to resistant osteosarcoma cells in the previous sections. Perhaps drug-resistant osteosarcoma cells are very similar to CSCs, so further understanding into the nature of CSCs and development of methods to target their unique features will be crucial in finding new treatment strategies against resistant osteosarcoma.

The tumour microenvironment can influence the efficacy of anticancer therapy (Figure 2I) [42]. In line with this notion, one study found that an acidic tumour microenvironment can cause multidrug resistance in osteosarcoma. Reducing the extracellular pH surrounding P-gp negative osteosarcoma cell lines from standard 7.4 to 6.5 reduced doxorubicin sensitivity, and the combination of doxorubicin with the proton pump inhibitor omeprazole significantly enhanced cytotoxicity. The combination of omeprazole and doxorubicin significantly reduced tumour volume and body weight loss in osteosarcoma xenografts. The impaired effect of doxorubicin under acidic conditions was attributed to a reversal of the pH gradient at the plasma membrane, which eventually led to reduced cellular accumulation. Reversing this pH gradient rendered cells sensitive not only to doxorubicin but also to cisplatin and methotrexate [283]. These findings suggest that multidrug resistance in osteosarcoma can be prevented by reducing the acidity of the tumour microenvironment, perhaps with the concurrent administration of proton pump inhibitors together with chemotherapy agents. A caveat of this strategy is the possibility of increased methotrexate toxicities [396].

Trabectedin is a chemotherapeutic agent that binds to DNA to cause damage and apoptosis that is used in the treatment of soft tissue sarcoma but has shown no clinical effect as a monotherapy in the treatment of osteosarcoma (see above). It has, however, been hypothesized to be useful in combination with other therapies in osteosarcoma due to its ability to suppress resistance-associated genes [141]. In a mouse osteosarcoma model, trabectedin was found to have the ability to suppress tumour growth and repress metastasis and to promote Runx-2 transcription factor binding in vivo and in vitro, most likely leading to osteogenic differentiation. Trabectedin led to the recruitment of CD8+ and CD4+ T cells to tumours in the more aggressive mouse model mOS69, which does not normally exhibit vast T cell infiltration, and increased expression of checkpoint inhibitor PD-1 on CD8+ cells, indicating T cell exhaustion. Treatment of mOS69 mice with trabectedin and an anti-PD-1 antibody resulted in significantly enhanced tumour inhibition [397]. These results demonstrate that trabectedin is able to reprogram the tumour microenvironment in osteosarcoma to recruit T cells that can be targeted with immune checkpoint inhibitors to overcome exhaustion. This is an exciting finding that can lead to new therapy options for patients with refractory or recurrent osteosarcoma.

### 5.10. Bone Pathways and Treatment Resistance

Cross-talk between the Hh, Notch and WNT bone development pathways may be involved in therapy resistance in osteosarcoma [21]. Increased activity of the Wnt/β-catenin signalling pathway has been observed in osteosarcoma [38,39], and a possible interaction between TRIM37 and β-catenin has been proposed to be involved in the aberrant signalling of this pathway in other cancer types [398]. High expression of *TRIM37* mRNA and protein was observed in paediatric osteosarcoma tumour samples. Treatment of cell lines Saos-2 and MG-63 with ifosfamide, doxorubicin, cisplatin, or methotrexate induced *TRIM37* expression. Upregulation of *TRIM37* in vitro induced drug resistance, whereas *TRIM37* knockdown restored chemosensitivity. The *TRIM37*-induced chemoresistance was found to be partially dependent on the activation of the Wnt/β-catenin signalling pathway [399]. These findings implicate the Wnt/β-catenin signalling pathway in chemoresistance in osteosarcoma and suggest that *TRIM37* could be used as a target to supress activity of this pathway and increase drug sensitivity. The expression of *NOTCH* genes has been associated with a more aggressive metastatic phenotype in osteosarcoma [36], and preclinical testing of the γ-secretase Notch pathway inhibitor RO4929097 inhibited growth in all six osteosarcoma xenografts tested [400]. A phase I and II clinical trial was launched combining Notch inhibition (using RO4929097) with Hh inhibition (using vismodegib) in patients with relapsed osteosarcoma (ClinicalTrials.gov Identifier NCT01154452). Unfortunately, this trial was prematurely closed due to discontinuation of RO4929097 manufacturing. It would be of interest to test the effect of this combination on relapsed osteosarcoma in the future.

Interestingly, Runx2 also functions as a growth suppressor and apoptosis inducer in normal osteoblasts. It has been found that its function as an inducer of osteoblast differentiation is perturbed in osteosarcoma [401]. Thus, enhanced drug sensitivity and apoptosis mediated by increased Runx2 expression observed in osteosarcoma cell lines [347] may be due to its role in bone regulatory pathways in addition to its role in apoptosis signals.

It is important to note, however, that targeting bone pathways might not be ideal in the treatment of resistant osteosarcoma in a paediatric population. These patients have not reached bone maturity, and perturbations in bone signalling could result in long-term side effects such as early-onset osteoporosis and growth defects. This therapy should, therefore, be evaluated in an adult population, and even then be considered with great caution due to the potential for skeletal side effects.

### 5.11. Other Factors Linked to Drug Resistance in Osteosarcoma

It has recently been found that expression of transferrin receptor-1 (TfR1) significantly correlates with poor survival, and that high expression of this factor was associated with a high histological grade, high Enneking staging, and metastases [375]. TfR1 is the main protein responsible for iron uptake, and abnormal iron metabolism is associated with tumorigenesis [402], suggesting that abnormal iron metabolism may be associated with chemotherapy resistance.

Whole-genome sequencing was done on two datasets of blood samples from patients receiving standard MAP neoadjuvant treatment (TARGET and ANOVA) in an attempt to reveal novel targets for personalized therapy. This revealed overlapping haplotypes associated with relapse. Among these were genes known to be linked to chemotherapy resistance, including AKR1D1 which is associated with drug metabolism and CDH13, CDH9, and PKHD1 that are part of a cell–cell adhesion process known to be associated with a multidrug-resistant phenotype in some tumours [403]. These findings need, however, prospective evaluation.

Overexpression of oestrogen-related receptor alpha (ERRα) was found to promote cell survival and inhibit methotrexate-induced cell death in U2-OS cells, which was associated with suppression of reactive oxygen species induction of p53-associated apoptosis pathways [404]. It was also reported that cisplatin- and doxorubicin-resistant osteosarcoma cell lines had high levels of ERRα and that targeted inhibition of ERRα by specific siRNAs or by the inverse agonist XCT-790 could restore chemosensitivity. Furthermore, ERRα was found to increase transcription of ABCB1, and XCT-790 decreased mRNA stability of ABCB1, suggesting that targeting ERRα could have potential in restoring chemotherapy resistance in osteosarcoma [405].

Overexpression of the circular RNA hsa-circ-0000073 in osteosarcoma cells was found to accelerate methotrexate resistance through inhibition of miR-145-5p and miR-151-3p-mediated downregulation of the cancer-related N-Ras pathway [406], opening up the potential that circular RNAs could be targets for future therapy. Comparison of microarray data from methotrexate-resistant and methotrexate-sensitive Saos-2 cell lines demonstrated that a subset of genes (AARS, AURKA, AURKB, CENPA, CCNB1, CCNE2, and CDK) that may contribute to methotrexate resistance were involved in aminoacyl-tRNA biosynthesis pathway, cell cycle pathway, or p53 signalling pathway [407].

## 6. Mechanisms Underlying Drug Efficacy/Synergy

Given the toxic side effects of many chemotherapeutic agents, it would be ideal to give these powerful agents at the lowest possible dose with the same, or better, effect. One way to achieve this is by addition of a factor that promotes drug synergy based on a known molecular mechanism. To do this, the molecular mechanism behind MAP synergy must be uncovered. A recent publication from our laboratory showed that drug synergy could be promoted in the treatment of acute myeloid leukaemia by inhibiting a known molecular factor that inhibits drug efficacy [408], and the same idea can be employed in osteosarcoma. Currently, our laboratory is making efforts to uncover the molecular basis of the synergy of conventional chemotherapy agents in several cell lines.

Another method of increasing drug efficacy is to alter signals in the tumour microenvironment. Osteosarcomas express high levels of the surface marker CD47, which prevents their engulfment and clearing by tumour-associated macrophages (TAMs), and blocking CD47 in osteosarcoma xenografts has been found to inhibit tumour progression by activating tumour cell clearance by TAMs [409]. The use of a monoclonal antibody against CD47 was not, however, able to entirely eradicate osteosarcoma in a mouse model [410], suggesting that it, like other antibody-based treatments of osteosarcoma, is not sufficient as a monotherapy. Doxorubicin has been found to induce immunogenic cell death by upregulating signals that encourage TAM-mediated cell clearance [411], suggesting that a combination of a monoclonal anti-CD47 antibody (CD47 mAb) and doxorubicin could increase drug efficacy in osteosarcoma. Accordingly, the combination of CD47 mAb and doxorubicin significantly reduced tumour size and prolonged survival in osteosarcoma tumour-bearing mice compared to animals receiving either monotherapy or control IgG. This was found to correlate with elevated TAM quantities, and it was presumed that increased tumour clearance was achieved by turning TAM-related “eat me” signals on (doxorubicin) and “do not eat me” signals off (CD47 mAb). [412]. This combination could be useful in increasing doxorubicin efficacy in a clinical setting as well. Finally, it has also been found that epigenetic reprogramming using inhibitors of DNA methyltransferases can induce differentiation of osteosarcoma cell lines into more differentiated cells [413], suggesting that epigenetics may be another method to increase therapy efficacy in osteosarcoma.

## 7. Conclusions

Even though osteosarcoma is considered a chemotherapy-resistant malignancy, intensive combination chemotherapy was able to substantially improve survival. Nevertheless, a plateau has been reached, and further therapy intensification has failed to improve outcomes since the 1980s. However, the long-term cure of 70% of patients certainly proves that osteosarcoma is susceptible to chemotherapy. We therefore believe that chemotherapy is, in principle, able to cure osteosarcoma, and hypothesise that specific mechanisms that confer chemoresistance can explain the failure to cure 30% of patients. Hence, targeting those resistance mechanisms promises to sensitise resistant osteosarcoma to standard therapies. As combination therapy is vastly more efficacious as compared to monotherapy, a special focus should be put on the identification of determinants of drug synergy. Those determinants might not only serve to stratify patients according to expected responses and inform ideal dosing and timing of combination chemotherapy, but also serve as therapeutic targets. Our laboratory is currently focusing on identifying genes that dictate synergy of standard osteosarcoma chemotherapeutics. Given the undebatable role of the tumour microenvironment and the anecdotal evidence of efficacy of immunotherapy, the effects of cytotoxic drugs on the tumour microenvironment and the immune system will have to be taken into account when incorporating immunomodulatory therapies into standard treatments. Finally, careful identification of subgroups of patients that might benefit from addition of targeted therapy to standard combination chemotherapy is another key for improving osteosarcoma survival. All of the above will hopefully allow us to break the spell of four decades of stagnation for osteosarcoma patients (Figure 3).

## Figures and Tables

**Figure 1 ijms-21-06885-f001:**
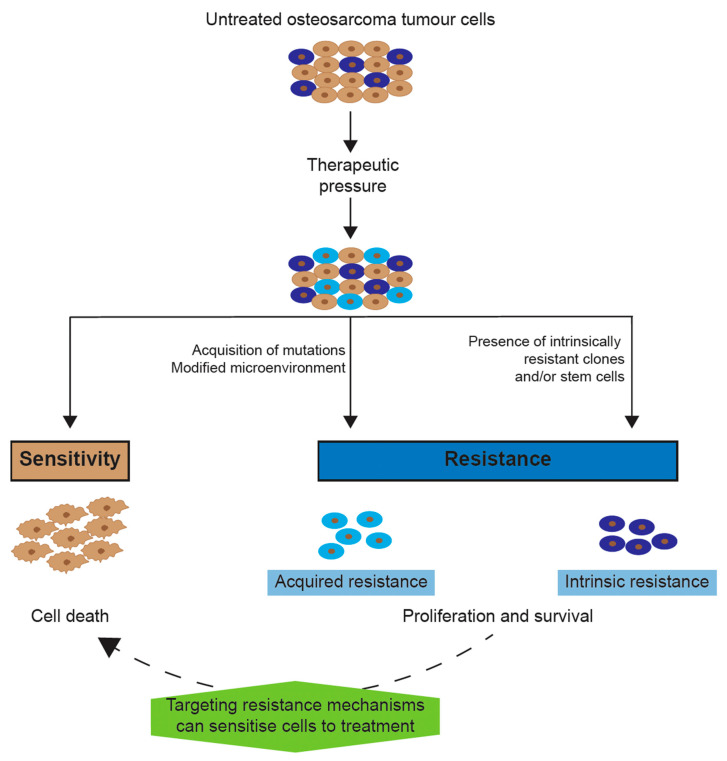
Model for chemotherapy resistance in osteosarcoma. The untreated population of tumour cells contains cells that are either sensitive (brown) to therapy or intrinsically resistant (purple). Therapy eradicates the sensitive population, while the cells with intrinsic resistance are able to survive along with cells that acquire resistance due to therapeutic pressure. The resistant populations can become sensitive to chemotherapy by targeting mechanisms underlying their resistance.

**Figure 2 ijms-21-06885-f002:**
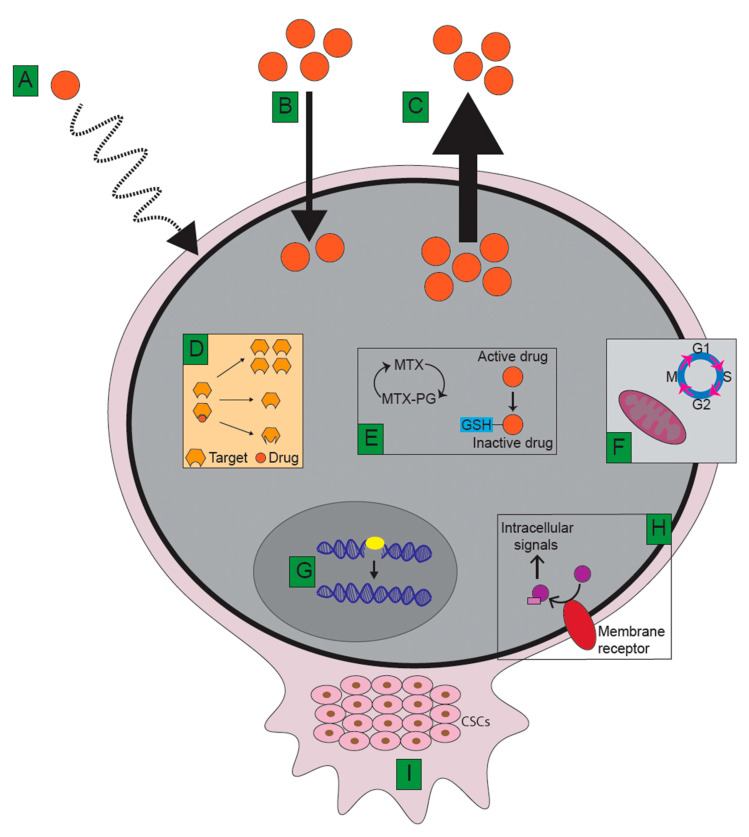
**Major mechanisms contributing to therapy resistance in an osteosarcoma tumour cell.** (**A**) Inefficient drug delivery and/or penetration. (**B**) Reduced drug influx due to alterations to drug carriers such as the reduced folate carrier. (**C**) Increased drug efflux due to upregulated ATP-binding cassette (ABC)-family transporters. **(D)** Reduced affinity of drug to its target due to target overexpression, repression, or mutations at the interaction site. (**E**) Reduced accumulation due to intracellular drug modifications such as changes in the polyglutamylation (PG) of methotrexate (MTX) or conjugation with glutathione (GSH). (**F**) Perturbations in apoptosis and cell cycle regulation. (**G**) Alterations in DNA repair pathways. (**H**) Altered signal transduction. (**I**) Cancer stem cells (CSCs) and tumour microenvironment.

**Figure 3 ijms-21-06885-f003:**
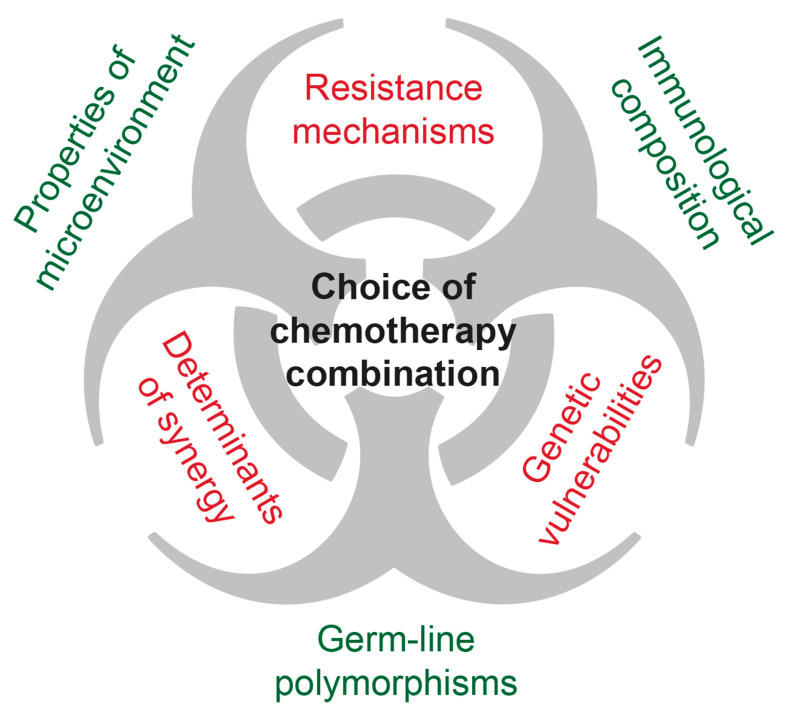
Schematic overview of somatic (red) and host (green) factors, which interact the efficacy of combination chemotherapy in osteosarcoma. Host factors: properties of microenvironment: vascularization, osteolytic activity, and inflammation are examples that modulate efficacy of chemotherapy. Immunological composition: the degree of immune cell infiltration, the type (e.g., tumour-associated macrophages and T-cells), and subtype (e.g., M2-polarization and CD8-positivity) as well as degree of immune cell exhaustion (e.g., expression of programmed death ligand-1 (PD-1)) correlate with treatment outcomes. Germ-line polymorphisms: pharmacokinetics of drugs like methotrexate modulate the effective drug exposure of tumour cells. Somatic factors: resistance mechanisms—upregulation and somatic mutations in, e.g., efflux pumps or catabolizing enzymes can detoxify osteosarcoma cells from cytotoxic drugs. Genetic vulnerabilities: activating mutations in signalling pathways can render osteosarcoma susceptible for small molecule inhibitors. Furthermore, while somatic defects in, e.g., the DNA damage response (“BRCAness”) might confer resistance to conventional cytotoxic agents, this also renders tumour cells vulnerable to synthetic lethality through poly (ADP-ribose) polymerase 1 (PARP) inhibitors. Determinants of synergy: currently unknown somatic factors that determine the extent of synergy, e.g., of cisplatin and doxorubicin, might help to predict responses to chemotherapy combinations. A rational choice of individualized combination chemotherapy would need to take these factors into account.

**Table 1 ijms-21-06885-t001:** Overview of novel therapies evaluated in clinical trials of osteosarcoma.

Targetable Elements	Function(s)	Evidence Found in Osteosarcoma Tissue Samples	Clinical Results in Human Trials
Unspecific immune modulators	Antiproliferative and proapoptotic effects in cancer cells	Inhibit tumour progression in preclinical models [197,198,200]	Pooled IFNα added to standard treatment yielded better outcome in a nonrandomized trial [194]
			IFN-α-2b together with MAP did not improve outcome in a large randomized trial [130]
			Inhalation of aerosolized GM-CSF did not hinder disease progression nor improve outcome [199]
			L-MTP-PE showed some effects on survival (but interactions with ifosfamide in a factorial study design) [116,201]
CTLA-4	Negative regulator of cytotoxic T-cells	Osteosarcomas are enriched in CTLA-4 pathways [52]	Ipilimumab showed partial responses or stable disease in a small number of paediatric patients [204]
PD-1	Negative regulator of cytotoxic T-cells	Patients positive for PD-1 ligand had worse event-free survival, and osteosarcomas are enriched in PD-1 signalling [52]	Nivolumab showed some efficacy in adults [205]
			Pembrolizumab had limited efficacy in adults [206]
ErbB/Her receptors (including Her1, Her2, Her3, Her4, and EGFR)	Involved in differentiation, proliferation, and cell cycle control	Overexpression of Her2/HER2 and EGFR in patient samples [93,94]	Trastuzumab (anti-Her2) in combination with MAP did not reduce metastatic disease nor improve overall outcome [229]
Receptor-associated and cytoplasmic protein kinases, including MAPK, BRAF, VEGFRs, PDGFR, and IGF-1R	Transmembrane signalling and intracellular transduction	Overexpression of several tyrosine kinases and their receptors in patients linked to poor response to therapy and outcome [29,103,104,105,107]	The broad tyrosine kinase inhibitor sorafenib led to a 46% progression-free survival after 4 months [241]
			Sorafenib plus the mTOR inhibitor everolimus did not improve 6-month progression-free survival [242]
			Regorafenib, a multikinase inhibitor with broader activity than sorafenib, led to reduced disease progression [245,246]
			Robatumumab, a monoclonal anti-IGF1R antibody, had little to no effect on clinical outcomes [266]
			Cabozantinib (VEGFR2 and MET inhibitor) yielded partial response and inhibited disease progression in some patients [248]
GpNMB	Involved in tissue repair, adhesion, and growth	gpNMB mRNA found to be overexpressed in primary patient samples [261]	Glembatumumab vedotin (gpNMB antibody coupled to a microtubule inhibitor) did not have significant clinical efficacy [264]
Small Rho GTPases	Promote osteoclast activity	Increased RhoA activity found in some metastatic osteosarcoma patients and found in osteosarcoma mice with metastatic disease [251]	Zoledronate (a bisphosphonate that inhibits Rho) did not improve outcome when combined with traditional therapy [253]

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
