# Peer review of "Targeting Molecular Mechanisms Underlying Treatment Efficacy and Resistance in Osteosarcoma: A Review of Current and Future Strategies"

_ijms, 2020, doi:10.3390/ijms21186885_

Round 1

Reviewer 1 Report

In this paper the authors try to summarize current osteosarcoma treatment strategies focusing on molecular determinants of treatment susceptibility and resistance.

This is an interesting, extensive and well written review that should be considered for publication.

The only point that can be raised is a sort of umbalance among the different sections, for example the molecular biology section, and in particular the “Cellular signalling pathways” subsection could be improved as far as in the last five years several new pathways were indagated. A more complete report of the last published experiences in this peculiar area could make the paper more appealing

Author Response

The authors would like to thank the reviewers for their thorough evaluation of the manuscript that has led to a substantial improvement of the original manuscript in the current revised version. We have tried to address all comments, and a detailed point-by-response can be found below (with our responses in italic red font). Furthermore, we have added an introductory sentence to the abstract (“Osteosarcoma is the most common primary malignant bone tumour in children and adolescents.”).

Reviewer 1

In this paper the authors try to summarize current osteosarcoma treatment strategies focusing on molecular determinants of treatment susceptibility and resistance.

This is an interesting, extensive and well written review that should be considered for publication.

The only point that can be raised is a sort of umbalance among the different sections, for example the molecular biology section, and in particular the “Cellular signalling pathways” subsection could be improved as far as in the last five years several new pathways were indagated. A more complete report of the last published experiences in this peculiar area could make the paper more appealing.

We would like to thank the reviewer for the relevant criticism. The revised manuscript has expanded the molecular biology section by adding pre-clinical and clinical data on the role of the Hippo pathway as well as TGF-b signaling in osteosarcoma. We think that this important addition has further improved the manuscript.

The following section has been added:

More recently, dysregulation of the tumour-suppressive Hippo pathway has been implicated in osteosarcoma [109]. The stem cell transcription factor Sox2, a supposed marker for cancer stem cells and highly expressed in osteosarcoma [110], downregulates the Hippo activators merlin (Nf2) and WW domain-containing protein 1 (WWC1) and upregulates the Hippo suppressor Yes-associated protein 1 (YAP) [109].

Following binding to its receptors, Transforming Growth Factor-β (TGF-β) signals mainly through Smad3/4 proteins that act as transcription factors following nuclear translocation in both osteosarcoma cells and its microenvironment, promoting osteolysis, angiogenesis, and metastases (for review, see [111]). Inhibiting TGF-β signaling in murine osteosarcoma models led to reduced lung metastasization [112].

Reviewer 2 Report

I'm just satisfied with the new version revised according to other reviewers.

Reviewer 3 Report

Comments to Authors

This is a very good review about the treatment evolution in osteosarcoma, as well as a very extensive overview of the experimental findings which may (hopefully) be translated into clinical practice to improve cure rate and prognosis of this tumor.

Authors must be acknowledged for having provided such an extensive overview of what has been done and the state of art of osteosarcoma treatment and I would like to congratulate with Authors for their excellent work.

The language is very clear, the cited literature is updated, and the Figures are explanatory and well organized (please, see the note for Figure 3 below).

I personally do not have specific requests for changes, but I would like to address a couple of non-mandatory suggestions to Authors in order to improve the already excellent quality of their work.

1) As a reader, I would appreciate a summary Table listing the non-conventional treatments that have been explored in high-grade osteosarcoma at clinical level and have provided results.

They are very well described in the text, but a summary Table would be very helpful.

I would suggest a Table with only the major characteristics of each target/tailored/non-conventional trial, without references in order to not be redundant with the text.

For example, I might suggest to organize the Table as follows:

Column 1: Biomarker(s)

(i.e. ErbB/HER receptors, including Her1, Her2, Her3, Her4, and EGFR)

Column 2: Function(s)

(i.e. Involved in differentiation, proliferation, and cell cycle control)

Column 3: Evidence found in osteosarcoma tissue samples

(i.e. Overexpression of Her2 protein and HER2 gene)

Column 4: Clinical results

(i.e. No improvement in patients' outcome after treatment with trastuzumab in addition to MAP therapy)

However, as stated above, this is only a non-mandatory suggestion.

2) In Figure 3, it is not clear to me what the Authors means with "Genetic vulnerabilities" and "Determinants of Synergy".

For example, I may suppose that "Genetic vulnerabilities" means "presence of genetic therapeutic targets", but maybe the Authors meant something else.

The use of alternative terms may be clearer for readers.

3) Please, note that the title of reference 364 (Fanelli et al, 2020) is incomplete. It has to be:

Cisplatin Resistance in Osteosarcoma: In vitro Validation of Candidate DNA Repair-Related Therapeutic Targets and Drugs for Tailored Treatments.

Author Response

The authors would like to thank the reviewers for their thorough evaluation of the manuscript that has led to a substantial improvement of the original manuscript in the current revised version. We have tried to address all comments, and a detailed point-by-response can be found below (with our responses in italic red font). Furthermore, we have added an introductory sentence to the abstract (“Osteosarcoma is the most common primary malignant bone tumour in children and adolescents.”).

Reviewer 2

This is a very good review about the treatment evolution in osteosarcoma, as well as a very extensive overview of the experimental findings which may (hopefully) be translated into clinical practice to improve cure rate and prognosis of this tumor.

Authors must be acknowledged for having provided such an extensive overview of what has been done and the state of art of osteosarcoma treatment and I would like to congratulate with Authors for their excellent work.

The language is very clear, the cited literature is updated, and the Figures are explanatory and well organized (please, see the note for Figure 3 below).

I personally do not have specific requests for changes, but I would like to address a couple of non-mandatory suggestions to Authors in order to improve the already excellent quality of their work.

 We would like to thank the reviewer for his/her appreciation of our manuscript.

1) As a reader, I would appreciate a summary Table listing the non-conventional treatments that have been explored in high-grade osteosarcoma at clinical level and have provided results.

They are very well described in the text, but a summary Table would be very helpful.

I would suggest a Table with only the major characteristics of each target/tailored/non-conventional trial, without references in order to not be redundant with the text. 

For example, I might suggest to organize the Table as follows: 

Column 1: Biomarker(s)

(i.e. ErbB/HER receptors, including Her1, Her2, Her3, Her4, and EGFR)  

Column 2: Function(s)

(i.e. Involved in differentiation, proliferation, and cell cycle control) 

Column 3: Evidence found in osteosarcoma tissue samples

(i.e. Overexpression of Her2 protein and HER2 gene)

 Column 4: Clinical results

(i.e. No improvement in patients' outcome after treatment with trastuzumab in addition to MAP therapy) 

However, as stated above, this is only a non-mandatory suggestion.

 We have now incorporated a table (Table 1) into the manuscript according to the reviewer’s suggestion. While assembling this table, we realised that bisphosphonates needed a more detailed work-up in the main text. Therefore, we have added the following paragraph:

Zoledronate is a nitrogen-containing bisphosphonate that has been shown to be effective in hindering osteosarcoma tumour progression and metastasis in mouse models (reviewed in [252]). Given these results, the OS2006 trial attempted to improve outcome in osteosarcoma patients by combining zoledronate with chemotherapy and surgery in a phase III clinical trial. The results were disappointing, however, with poorer event-free 3-year survival observed in the zoledronate group compared with the control group (57.1% vs 63.4%) [253]. This suggests that bisphosphonates, despite promising preclinical evidence, may not be of benefit in osteosarcoma patients, and further studies are needed to investigate the reason for this discordance.

 2) In Figure 3, it is not clear to me what the Authors means with "Genetic vulnerabilities" and "Determinants of Synergy".

For example, I may suppose that "Genetic vulnerabilities" means "presence of genetic therapeutic targets", but maybe the Authors meant something else.

The use of alternative terms may be clearer for readers.

 We would like to thank the reviewer for pointing out a lack of clarity in Figure 3. We have now added a more exhaustive figure legend that explains the individual terms, in particular "Genetic vulnerabilities" and "Determinants of Synergy". The revised figure legend reads as follows:

Figure 3. Schematic overview of somatic (red) and host (green) factors which interact the efficacy of combination chemotherapy in osteosarcoma. Host factors: Properties of microenvironment: vascularization, osteolytic activity, and inflammation are examples that modulate efficacy of chemotherapy. Immunological composition: The degree of immune cell infiltration, the type (e.g. tumor-associated macrophages and T-cells) and subtype (e.g. M2-polarization and CD8-positivity) as well as degree of immune cell exhaustion (e.g. expression of PD-1) correlate with treatment outcomes. Germ-line polymorphisms: Pharmacokinetics of drugs like methotrexate modulate the effective drug exposure of tumor cells. Somatic factors: Resistance mechanisms: Upregulation and somatic mutations in e.g. efflux pumps or catabolising enzymes can detoxify osteosarcoma cells from cytotoxic drugs. Genetic vulnerabilities: Activating mutations in signaling pathways can render osteosarcoma susceptible for small molecule inhibitors. Furthermore, while somatic defects in e.g. the DNA damage response (“BRCAness”) might confer resistance to conventional cytotoxic agents, this also renders tumor cells vulnerable to synthetic lethality through PARP inhibitors. Determinants of synergy: Currently unknown somatic factors that determine the extent of synergy, e.g. of cisplatin and doxorubicin, might help to predict responses to chemotherapy combinations. A rational choice of individualised combination chemotherapy would need to take these factors into account.

 3) Please, note that the title of reference 364 (Fanelli et al, 2020) is incomplete. It has to be:

Cisplatin Resistance in Osteosarcoma: In vitro Validation of Candidate DNA Repair-Related Therapeutic Targets and Drugs for Tailored Treatments.

This has been corrected.
